# Inhibition of tetrameric Patched1 by Sonic Hedgehog through an asymmetric paradigm

Hongwu Qian[1,4], Pingping Cao [2,4], Miaohui Hu[1], Shuai Gao [1], Nieng Yan[1] & Xin Gong[1,3]

The Hedgehog (Hh) pathway controls embryonic development and postnatal tissue maintenance and regeneration. Inhibition of Hh receptor Patched (Ptch) by the Hh ligands relieves suppression of signaling cascades. Here, we report the cryo-EM structure of tetrameric Ptch1 in complex with the palmitoylated N-terminal signaling domain of human Sonic hedgehog (ShhN$_p$) at a 4:2 stoichiometric ratio. The structure shows that four Ptch1 protomers are organized as a loose dimer of dimers. Each dimer binds to one ShhN$_p$ through two distinct inhibitory interfaces, one mainly through the N-terminal peptide and the palmitoyl moiety of ShhN$_p$ and the other through the $Ca^{2+}$-mediated interface on ShhN$_p$. Map comparison reveals that the cholesteryl moiety of native ShhN occupies a recently identified extracellular steroid binding pocket in Ptch1. Our structure elucidates the tetrameric assembly of Ptch1 and suggests an asymmetric mode of action of the Hh ligands for inhibiting the potential cholesterol transport activity of Ptch1.

[1] Department of Molecular Biology, Princeton University, Princeton, NJ 08544, USA. [2] State Key Laboratory of Membrane Biology, Beijing Advanced Innovation Center for Structural Biology, Tsinghua-Peking Joint Center for Life Sciences, School of Life Sciences and School of Medicine, Tsinghua University, Beijing 100084, China. [3] Department of Biology, Southern University of Science and Technology, Shenzhen, Guangdong 518055, China. [4]These authors contributed equally: Hongwu Qian, Pingping Cao. Correspondence and requests for materials should be addressed to N.Y. (email: nyan@princeton.edu) or to X.G. (email: gongx@sustech.edu.cn)

The Hedgehog (Hh) precursor undergoes autocatalytic processing and the resulting N-terminal signaling domain is modified by N-terminal palmitoylation and C-terminal cholesterylation[1–3]. Binding of the processed Hh ligand to the surface receptor Patched (Ptch) relieves suppression of the downstream G-protein coupled receptor, Smoothened (Smo). This subsequently activates the Hh signaling cascade that controls embryogenesis and tissue homeostasis[4–6]. Deficient Hh signaling can lead to birth defects, whereas abnormal activation of Hh signaling is associated with various human cancers[7–9]. Small molecule modulators have been identified to target this pathway, and inhibitors of the Hh signaling have been explored as potential therapeutics for cancer treatment[10,11].

Due to the lack of a direct interaction between the Hh receptor Ptch and the downstream Smo[12–14], Ptch was suggested to inhibit Smo through an indirect mechanism, possibly by limiting the availability of modulatory ligand(s) to Smo[15]. Ptch shares sequence homology with the prokaryotic resistance-nodulation-division (RND) family transporters, exemplified by the bacterial proton-driven multidrug resistance exporter AcrB[16]. This phylogenetic evidence led to a model suggesting that Ptch may act as a transporter for the ligands, antagonists or agonists, of Smo.

The physiological importance of Ptch is underscored by its association with basal cell carcinoma, the most common cancer in humans[17,18]. The structure of Ptch1 was recently reported by several groups, including ours[19–21]. The 1447 amino acids in human Ptch1 fold to a tripartite architecture containing two extracellular domains (ECD1 and ECD2), a 12- transmembrane helix (TM) transmembrane domain (TMD) that exhibits a two-fold pseudo-symmetry, and intracellular domains that were unresolved in all the reported structures.

Distinct from most bacterial RND transporters whose structures were resolved as trimers[22], all the physiologically relevant Ptch1 structures were in monomeric form[19,21]. It has been reported that the intracellular middle-loop domain (MLD) and C-terminal domain (CTD) of Ptch1 mediated the oligomerization of Ptch1[23] and the CTD of *Drosophila* Ptc formed a trimer[24]. In contrast, another mammalian RND homolog, human Niemann-Pick type C1 (NPC1), appears to be a monomer[25,26]. Therefore, the oligomeric state of Ptch1 requires further investigation.

In our cryo-EM structure of the monomeric human Ptch1 (the C-terminal half of CTD truncated) in complex with an unmodified N-terminal domain of Sonic Hh (ShhN), ShhN engages its concave side to bind to Ptch1 through extensive polar interactions. Ensuing biochemical and structural characterizations revealed that formation of this interface requires binding of steroid to both the sterol-sensing domain (SSD) and an extracellular steroid binding site (ESBS), enclosed by ECD1 and ECD2[19]. We suggested that ShhN may relieve Smo inhibition by preventing conformational changes of Ptch1 that are required for its transport activity. Two weeks later, Qi et al. published the structure of native lipid-modified ShhN (hereafter designated ShhN$_n$) bound to a monomeric mutant Ptch1* (both MLD and CTD truncated)[21]. While the N-terminal palmitoyl moiety and the ensuing fragment of ShhN$_n$ bind to a pocket enclosed by ECD and TMD of Ptch1*, the globular protein domain of ShhN$_n$ only has very limited contact to Ptch1* through its convex side. Considering the oligomerization of Ptch1, the binding and inhibition of oligomeric Ptch1 by ShhN is thereby more complex and remains to be investigated. Here, we report the cryo-EM structure of tetrameric Ptch1 in complex with the palmitoylated ShhN (ShhN$_p$) at a 4:2 stoichiometric ratio.

## Results

**Purification of the Ptch1 and palmitoylated ShhN complex.** We have obtained an optimal human Ptch1 construct (residues 1–1305) that had markedly improved overexpression level and solution behavior compared to the full-length Ptch1 (Supplementary Fig. 1a)[19]. The major species of this Ptch1 construct existed in an oligomeric form upon size-exclusion chromatography (SEC) (Supplementary Fig. 1b), although the minor monomeric form was used for cryo-EM analysis in our previous study. Cryo-samples made from the oligomeric peaks were highly heterogeneous, impeding structural determination to high resolution[19]. Several attempts were made to overcome the heterogeneity of oligomeric Ptch1, including screening of detergents and buffer conditions, engineering of protein with distinct boundaries, and chemical cross-linking. The cryo-sample became amenable for cryo-EM analysis when glycol-diosgenin (GDN, Anatrace) was used for protein extraction and purification.

Details for protein generation can be found in the Methods section. Briefly, the human Ptch1 (residues 1–1305), with an N-terminal FLAG tag and a C-terminal His$_{10}$ tag, was co-expressed with untagged human ShhN (residues 1–197) in human embryonic kidney (HEK) 293F cells. After tandem affinity purification, the complex was eluted from SEC mainly in an oligomeric form with an elution volume similar to Ptch1 oligomer, and migrated at similar position as Ptch1 oligomer on blue native PAGE (BN-PAGE). The oligomeric Ptch1 and Ptch1/ShhN$_p$ complex migrated as a single band on BN-PAGE, suggesting the oligomers with uniform stoichiometry (Fig. 1a, Supplementary Fig. 1b). We expected the removal of the signal peptide of ShhN expressed in HEK 293F cells, and the resulting segment containing residues 24–197 to be palmitoylated but without C-terminal cholesterylation[3]. The mass spectrometric analysis confirmed the palmitoylation at Cys24 of ShhN in the co-expressed complex. We will refer to this protein as ShhN$_p$.

To characterize the oligomeric states of Ptch1 in solution, we performed glutaraldehyde-mediated cross-linking experiments using purified proteins. The increase in glutaraldehyde concentrations leads to the formation of crosslinked oligomers, which seems to be dimers and tetramers in SDS-PAGE (Supplementary Fig. 1c). The sedimentation velocity analytical ultracentrifugation (AUC-SV) measured the molecular weight of Ptch1/ShhN$_p$ complex was about 565 kDa (Supplementary Fig. 1d; the theoretical MW of Ptch1 and ShhN$_p$ was 145 kDa and 20 kDa, respectively). Taken together, the BN-PAGE, cross-linking and AUC-SV results supported the tetrameric assembly of Ptch1 both alone and in complex with ShhN$_p$.

**Structural determination of the 4:2 Ptch1–ShhN$_p$ complex.** Protocols for grid preparation, cryo-EM data acquisition, and structural determination of oligomeric Ptch1 in complex with ShhN$_p$ are described in detail in Fig. 1b, Supplementary Figs. 2, 3, and Methods. After 3D classification, two major maps were obtained at 6.8 Å and 6.5 Å resolutions out of 39,503 and 25,510 selected particles, respectively (Fig. 1c, Supplementary Figs. 2 and 3). The resolutions were sufficient to resolve the secondary structure features for the majority of ECD and TMD (Supplementary Fig. 3e). Some amorphous densities were observed on the intracellular side of the detergent micelles, likely belonging to the intracellular domains (Fig. 1c). After application of an adapted mask to ECDs, the resolutions of ECDs in the two maps were increased to 4.6 Å and 4.3 Å, respectively (Supplementary Figs. 2 and 3b).

The two 3D reconstructions of Ptch1–ShhN$_p$ complex revealed a similar 4:2 assembly (Fig. 1c). The four Ptch1 molecules were organized in two slightly different tetrameric forms, although

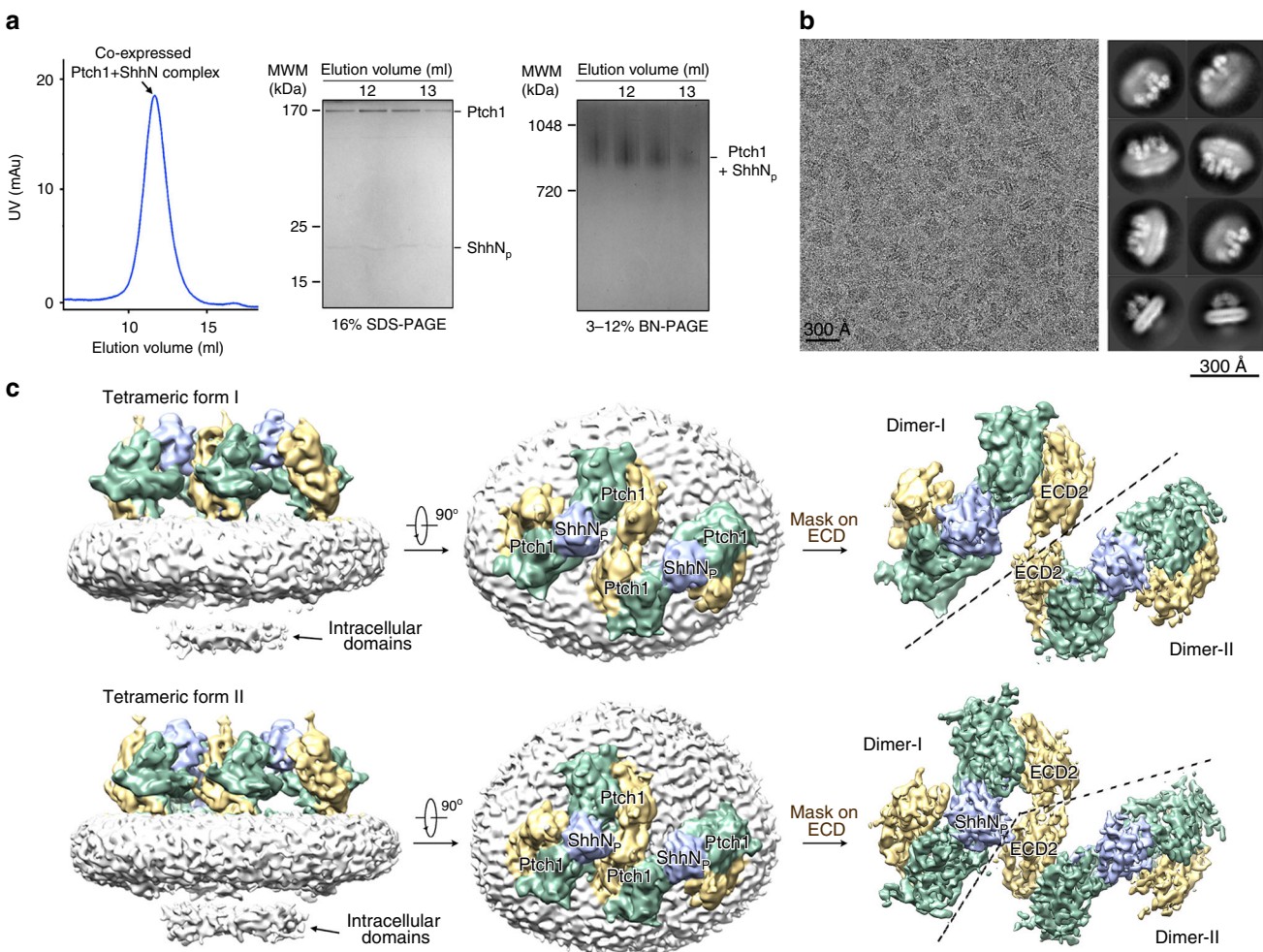

**Fig. 1** Structural determination of tetrameric Ptch1 in complex with ShhN$_p$. **a** Purification of co-expressed Ptch1 (residues 1–1305) and ShhN (residues 1–197). Shown here is a representative size exclusion chromatography (SEC) of the complex. The peak fractions were subjected to SDS-PAGE and blue native (BN)-PAGE and visualized by Coomassie-blue staining. The apparent molecular weight in BN-PAGE contains the surrounding detergent micelles. Source data are provided as a Source Data file. **b** Representative cryo-EM micrograph and 2D class averages. **c** Two three-dimensional reconstructions of the complex. Side and extracellular views are shown for both reconstructions. With adapted mask on extracellular domain (ECD), improved density maps for the ECD are shown on the right. The maps were generated in Chimera[61]

each form is a dimer of dimers. One ShhN$_p$ molecule is positioned in the middle of each dimer and simultaneously interacts with the extracellular domains from two Ptch1 protomers (Fig. 1c). The two dimers have a relative rotation along an axis perpendicular to the membrane in the two distinct tetrameric assemblies (Fig. 1c and Supplementary Fig. 3d). The two dimers loosely contact each other through ECD2 in both tetrameric assemblies, without any interaction in the TMDs (Fig. 1c and Supplementary Fig. 3c). The weak interactions may lead to structural flexibilities that limited the resolution of the cryo-EM reconstruction of the tetramer.

**Structure of the 2:1 Ptch1–ShhN$_p$ complex.** Considering that Ptch1 dimer with one ShhN$_p$ appears to be the basic unit within each 4:2 complex, we further applied adapted mask on one dimer to reduce the heterogeneity caused by the distinct dimer of dimers. The resulting EM map for the 2:1 Ptch1–ShhN$_p$ complex, out of 171,590 selected particles, reached the resolution of 3.6 Å according to the gold-standard Fourier shell correlation (FSC) 0.143 criterion (Fig. 2a, b and Supplementary Fig. 2). The map was well resolved for the majority of two Ptch1 protomers and one ShhN$_p$ molecule (Fig. 2a and Supplementary Fig. 4). A total

of 2153 residues were built with 2110 side chains assigned (Supplementary Table 1). The intracellular segments remained poorly resolved likely due to their flexibility. Consistent with the previous report that the intracellular domains are responsible for the oligomerization of Ptch1[23], the two Ptch1 protomers had no contact in the resolved structure (Fig. 2c).

One ShhN$_p$ simultaneously recognizes two Ptch1 through distinct interfaces (Fig. 2c). To facilitate illustration, we named the two Ptch1 protomers G (for interaction with the globular domain) and P (for interaction with the palmitoylate and peptide). Ptch1-G interacts with the pseudo-active site groove (the Ca$^{2+}$-mediated interface) of ShhN$_p$, corresponding to the complex resolved by us[19]. Ptch1-P mainly accommodates the N-terminal fragment and palmitoyl moiety with limited contact with the globular domain of ShhN$_p$, corresponding to the one in the Ptch1*–ShhN$_n$ complex[21] (Fig. 2d). The transmembrane domains of the two Ptch1 protomers stand in parallel within the membrane plane, with the SSDs sandwiched inside (Fig. 2e).

**The asymmetric binding of one ShhN$_p$ with two Ptch1.** Superimposing the structures of the 2:1 Ptch1–ShhN$_p$ with our previous 1:1 Ptch1–ShhN (PDB code 6DMY), relative to Ptch1-G,

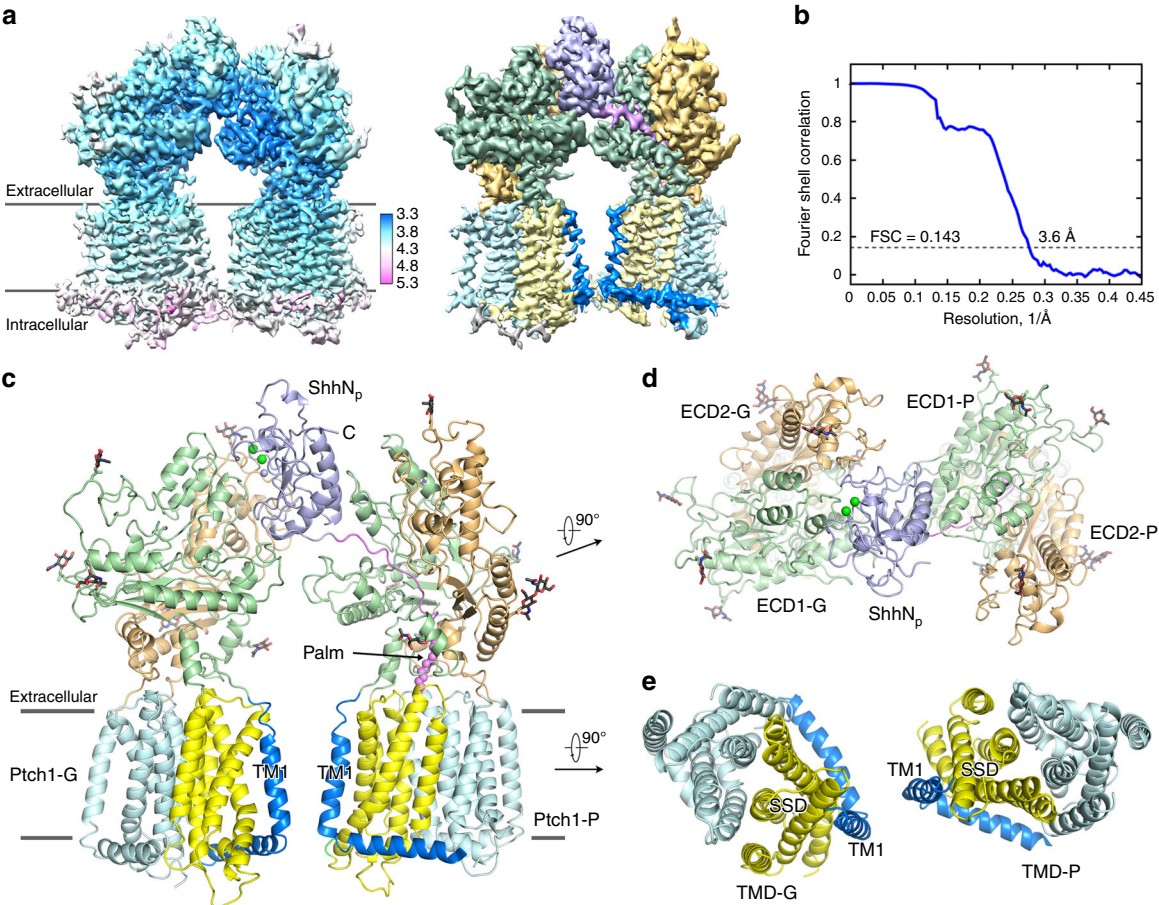

**Fig. 2** Structure of the 2:1 Ptch1–ShhNp complex. **a** Cryo-EM map for the 2:1 Ptch1–ShhNp complex. Higher resolution was achieved after applying adapted mask to dimeric Ptch1 with one ShhNp. Left panel: local resolution map calculated using RELION 2.0. Right panel: domain-colored map with the same color code for our previous report[19]. **b** The gold-standard Fourier shell correlation curve for the cryo-EM map. **c** Overall structure of one ShhNp bound to two Ptch1. To facilitate illustration, we named the two Ptch1 protomers as Ptch1-G (G for globular) and Ptch1-P (P for palmitoylate and peptide). The N-terminal palmitoyl moiety is highlighted as violet sphere. The $Ca^{2+}$ and $Zn^{2+}$ ions are shown as green and purple spheres, respectively. The glycosyl moieties are shown as black sticks. **d** ECDs and ShhNp viewed from extracellular view. **e** Transmembrane domains are shown in the extracellular view. All structure figures were prepared in PyMol[62]

reveals nearly no change in Ptch1 (Fig. 3a). The interface between ShhNp and Ptch1-G remains nearly identical to what we have described previously[19], hence will no longer be illustrated here.

Structural superimposition of our 2:1 complex with Ptch1*–ShhNn (PDB code 6D4J) revealed nearly identical structures for Ptch1, while ShhNp moved slightly away from Ptch1-P, likely due to its interaction with Ptch1-G (Supplementary Fig. 5). Examination of the deposited EM map for Ptch1*–ShhNn (EMDB code EMD-7796) showed that the local resolution for the interface mediated by the N-terminal segment of ShhNn and Ptch1* was ~5 Å, unable to support reliable analysis for detailed interactions. The local resolution in our current EM map enabled detailed analysis of the interactions between Ptch1-P and the N-terminal segment of ShhNp (Fig. 3b).

The N-terminal peptide (N15: $_{24}$CGPGRGFGKRRHPKK$_{38}$) and the palmitoyl moiety of ShhNp insert into a tunnel above TMD that is enclosed by ECD1 and ECD2 of Ptch1-P (Fig. 3b, c). The N15 segment mainly interacts with ECD1 of Ptch1-P through extensive polar interactions (Fig. 3b). The positively charged Cardin-Weintraub (CW) motif (residues 32–38) of ShhNp, which is important for binding to heparan sulfate proteoglycans (HSPGs) and Hh transport[27], dominates the polar interactions with Ptch1 (Fig. 3b). The palmitate binding pocket is formed by a number of hydrophobic residues from the TMD-

ECD connecting elements, including the Linker1, Neck helix 2, Linker7 and Neck helix 8, and TMs 4/6/10/12 (Fig. 3c). The limited contact between the convex side of ShhNp and Ptch1-P involves the E loop and ensuing helix α3 on ECD1, in part overlapping with the interface between Ptch1-G and ShhNp (Fig. 3d).

**The cholesteryl moiety of native ShhN**. During the preparation of this manuscript, a structure of monomeric Ptch1* in complex with ShhNn at 2:1 stoichiometric ratio was published[28]. Although Ptch1* is a monomeric mutant, one ShhNn can still bind to two Ptch1*. Structural comparison of the two 2:1 complexes reveals nearly identical architecture, except the two Ptch1* molecules move towards each other, relative to the ones in our Ptch1–ShhNp complex (Supplementary Fig. 6), possibly owing to the lack of stabilization by the intracellular domains in Ptch1*.

When the maps of the two 2:1 complexes were scrutinized, an extra EM density was found in the Ptch1*-ShhNn complex (EMDB code EMD-8955) (Fig. 4a). This density, which was not structurally assigned by the authors, is contiguous with the C-terminus of ShhNn and protrudes into a binding cavity corresponding to the ESBS of Ptch1*-P (Fig. 4a). We tentatively built a structural model for this C-terminal segment

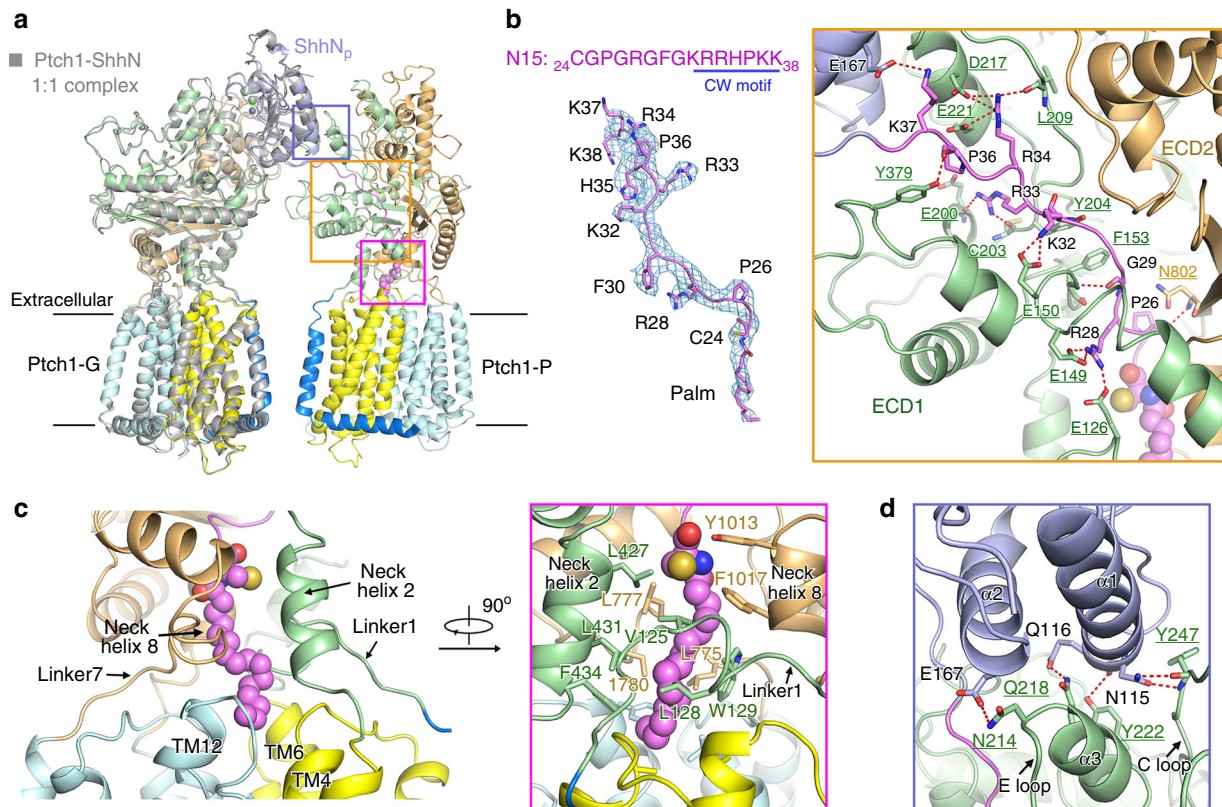

**Fig. 3** Asymmetric binding of one ShhN$_p$ with two Ptch1. **a** Structural comparison of 2:1 Ptch1–ShhN$_p$ with our previously reported 1:1 Ptch1–ShhN complexes. The 2:1 Ptch1–ShhN$_p$ complex was domain colored. The 1:1 Ptch1–ShhN complex (PDB code 6DMY) was colored gray. The superimposition was made relative to Ptch1-G molecule. **b** The N-terminal peptide (N15) and palmitoyl moiety of ShhN$_p$, which are well resolved in the 3.6 Å EM map for the 2:1 complex, interact extensively with ECD1 of Ptch1-P. The map, shown as blue mesh, was contoured at 6 σ. The potential electrostatic interactions are represented by dashed red lines. **c** The palmitate binding pocket in Ptch1-P. The hydrophobic residues in TMD-ECD connecting elements (including the Linker1, Neck helix 2, Linker7, and Neck helix 8) and hydrophobic surface residues in TM4/6/10/12 together constitute the binding pocket for the palmitate. The coordinating residues are shown as sticks and the palmitoyl moiety is shown as violet sphere. **d** The secondary interface between ShhN$_p$ and Ptch1-P. The globular domain of ShhN$_p$ interacts with ECD1 of Ptch1-P through limited polar interactions

(C7: $_{191}$VAAKSGG$_{197}$) and it became evident that the end density likely belongs to the cholesteryl moiety of ShhN$_n$ (Fig. 4b). The cholesteryl moiety is embedded in a cavity mainly enclosed by ECD1, where nearly 20 hydrophobic residues of ECD1 form the contour of the pocket (Fig. 4c). In support of the cholesteryl moiety binding to the ESBS, two cholesterol-like densities were also observed in the similar positions of two ESBS in our structure (Supplementary Fig. 4e)

**Different oligomeric states of Ptch1 and RND transporters.** The twelve TMs of Ptch1 protomer exhibit identical fold to those in the bacterial RND transporters, although their oligomeric organizations are different (Supplementary Fig. 7). Both the TMDs and ECDs of trimeric AcrB, the best characterized bacterial RND transporter, have extensive inter-protomer interactions to support trimer formation[29]. Such organization provides structural basis for the coupled rotating mechanism among the three protomers during substrate transport[30,31]. In contrast, the tetrameric assembly of Ptch1 only contains limited interaction between the ECDs of the dimer of dimers (Supplementary Figs. 7 and 8a, b). A variant containing mutations (△893-899/T903A/ K904A/Q905A/N1000Q) that alter the residues mediating the limited dimer-dimer interface on the extracellular side failed to disrupt the tetrameric assembly of Ptch1 alone or in complex with ShhNp (Supplementary Fig. 8c). The result was as expected because the tetrameric assembly is mainly mediated by the

intracellular domains, deletion of which resulted in the monomeric of Ptch1 both in vivo and in vitro[21,23].

Such architectural differences suggest that Ptch1 may not undergo a rotating mechanism observed for the bacterial RND transporters. In contrast, the four Ptch1 protomers likely operate independently of each other. Supporting this notion, the intracellular domain-deleted monomeric Ptch1 remained functional in cultured cell signaling assays[23].

## Discussion

Cholesterol was identified as an endogenous Smo agonist that can induce an active conformation of Smo[32–34]. A previous study has suggested that Ptch may act as a cholesterol exporter as a means to suppress Smo[35]. Consistent with this speculation, five cholesterol-like densities were observed in our current EM reconstruction: four occupying the previously identified binding sites on SSD and ESBS in both Ptch1 protomers and an extra one between SSD and ESBS in Ptch1-G. The corresponding site in Ptch1-P is occupied by the palmitoyl moiety (Supplementary Fig. 9a). A tunnel connecting the SSD and ESBS was observed in our previous structure[19]. In the present 2:1 complex, a similar tunnel stretches throughout the ECDs of Ptch1-G, but not Ptch1-P, that may represent the cholesterol transport path (Supplementary Fig. 9b). Similar sterol-like densities and a hydrophobic tunnel were also observed in an apo dimeric Ptch1 structure. Further biochemical evidence suggested that Ptch1 may mediate

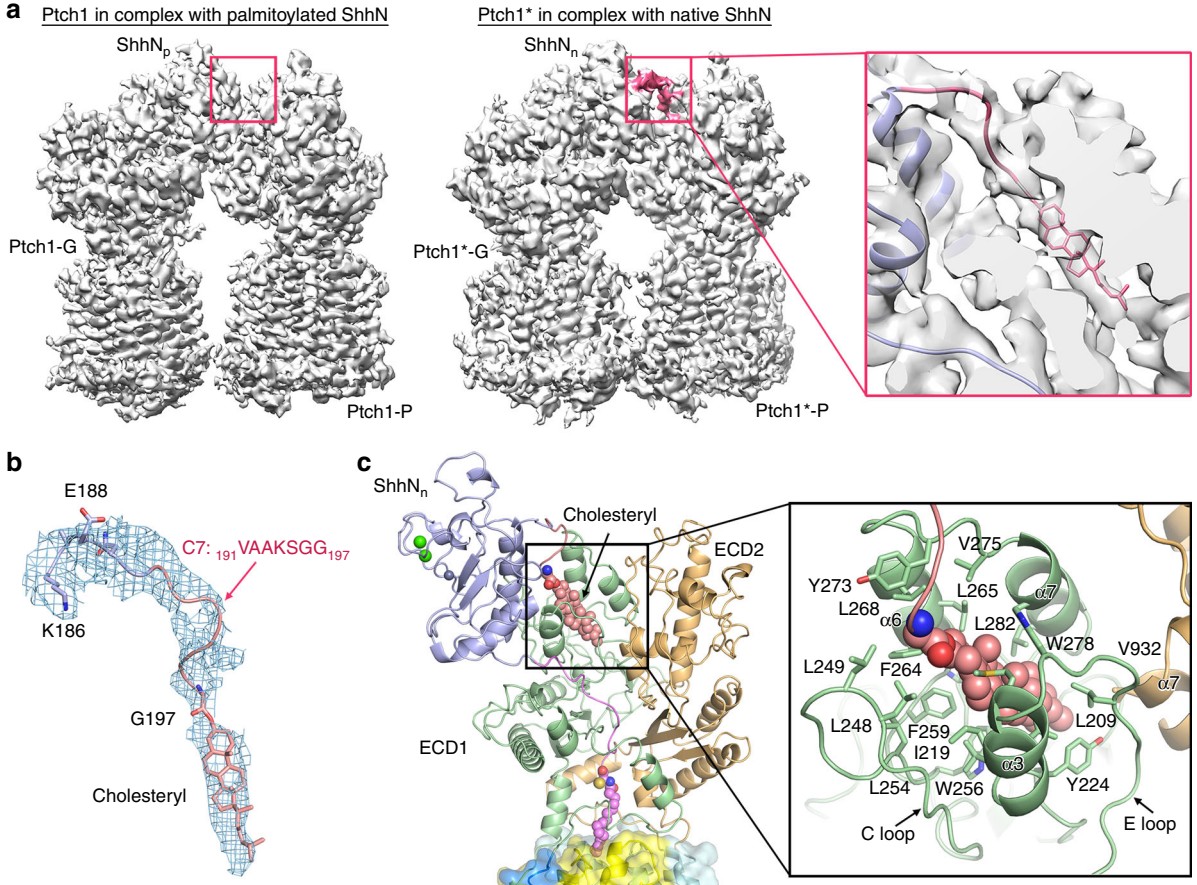

**Fig. 4** The cholesteryl moiety of native ShhN (ShhN_n). **a** Comparison of the EM maps for our Ptch1–ShhN_p and the recently published Ptch1*–ShhN_n (EMDB code EMD-8955) reveals an extra density of ShhN_n in the Ptch1*–ShhN_n map that protrudes into the extracellular steroid binding site (ESBS) of Ptch1*-P. The extra density is highlighted in pink. A zoomed-up cut-open view of the C-terminus of ShhN_n (ribbon representation) into ECD1 is shown on the right. **b** Density corresponding to the C-terminal peptide (C7) and cholesteryl moiety of ShhN_n contoured at 3 σ. **c**, The accommodation of the cholesteryl moiety by ESBS. As described previously, ESBS is mainly constituted by helices α3/α6/α7, E loop and C loop of ECD1. The hydrophobic residues that constitute the pocket are shown as sticks. The cholesteryl moiety is shown as pink sphere

the removal of cholesterol from the inner leaflet of the membrane to suppress Smo[20]. Nevertheless, the field still lacks a definitive experiment to directly demonstrate the cholesterol transport activity by Ptch1. In addition, an unidentified extracellular cholesterol carrier may be required to take the cholesterol expelled by Ptch1.

The asymmetric binding interfaces between one ShhN_p and two Ptch1 reveal two distinct inhibitory mechanisms for the two Ptch1 protomers by one ShhN_p. In both cases, ShhN binding would block the cholesterol transport activity of Ptch1. Inhibition of Ptch1-G was mediated by the globular domain of ShhN_p through a Ca²⁺-mediated interface. We previously proposed that ShhN binding to ECD1 and ECD2 of Ptch1 could prevent the conformational changes of Ptch1 required for its transport activity[19]. Inhibition of Ptch1-P was mainly mediated by the palmitoylated N15 peptide, which blocks the tunnel connecting SSD and ESBS (Supplementary Fig. 9b). The structure provides a nice explanation for the report that a palmitoylated N-terminal 22-residue peptide could partially activate Hh signaling by binding to Ptch1[36]. No substantial conformational changes were observed between the TMDs of Ptch1 alone and the two Ptch1 molecules in Ptch1–ShhN_p complex (Supplementary Fig. 9c), suggesting that ShhN_p association does not affect cholesterol binding to TMD.

An unexpected discovery is the insertion of the cholesteryl moiety into ESBS of Ptch1-P, which will naturally block binding

of cholesterol to this site, further blocking the transport of cholesterol along this path. Supporting this structural observation, cholesterol modification was shown to enhance the potency of signaling activity of Shh ligand in a Shh Light II cell-based report assay[37]. Nonetheless, the function of the cholesterylation remains to be further investigated. Contradictory observations of decreased, increased, or no change of the Hh activation as a result of the cholesterylation have been reported, probably owing to the distinct species and tissues examined[38–40].

The palmitoyl and cholesteryl moieties render the Hh ligands as hydrophobic morphogens; several factors have been described as Hh chaperones to ensure its solubility, such as Hh itself to form soluble multimeric Hh[41,42], lipoprotein particles[43,44], and Scube proteins[37,45]. Another Hh solubilization model involves the proteolytic shedding of membrane-associated Hh by a protease, resulting in the removal of both lipid moieties[46]. In this model, the N-palmitoylation is required for the proteolytic removal during solubilization[46,47]. Adding to these solubilization models, our structures suggest that Ptch1 binding could shield both the hydrophobic lipid moieties of ShhN from the aqueous environment (Fig. 5). It is also noted that the palmitoylation of Hh may not be required, but only restricted to specific species or tissues for signaling activation[48,49]. Taken together, these observations imply that the functional importance of lipid modifications of Hh is complicated. Our structural studies provide an alternative mechanism that awaits further investigations.

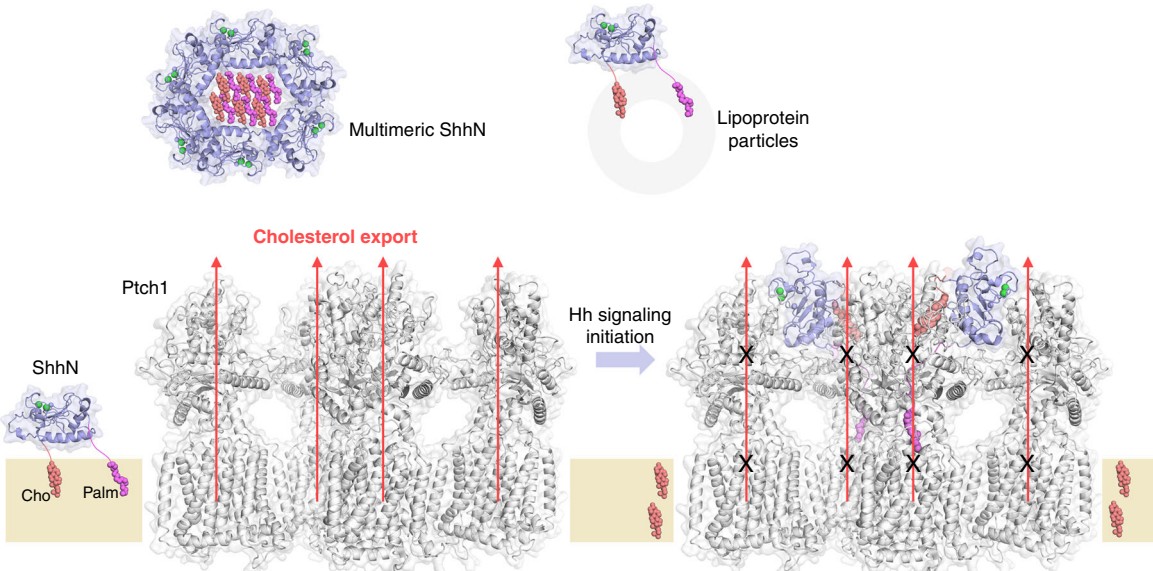

**Fig. 5** Model for the inhibition of tetrameric Ptch1 by lipid-modified ShhN. The palmitoyl and cholesteryl modified ShhN signaling domain becomes hydrophobic that can be attached to the membrane. Multimerization of the modified ShhN or interaction with lipoprotein particles keep the morphogen soluble. In the absence of Hh ligands, Ptch1 may export the cholesterol, the agonist to Smo, out of the lipid bilayer as a means to suppress Smo. Lipid-modified ShhN inhibits the cholesterol transport activity of Ptch1 using two asymmetric modes, resulting in elevated cholesterol in the membrane that leads to Smo activation

In summary, the structure of tetrameric Ptch1 in complex with palmitoylated ShhN at a 4:2 stoichiometric ratio further completes the molecular picture for the interplay between Ptch1 and Shh, setting an important framework for future investigation of Hh signaling (Fig. 5). ShhN binding can result in both Ptch1 inhibition and internalization from the primary cilia[14]. The tetrameric assembly of Ptch1 may increase the internalization efficiency or be prerequisite for internalization, a caveat to be investigated. Considering the oligomerization of Hh ligands, this could result in even higher-order clustering of Ptch1 at the cell surface[50]. It is noted that the asymmetric binding between the surface receptor and the ligand at a 2:1 stoichiometric ratio has been discovered in several cases, such as growth factor receptor[51], erythropoietin receptor[52], and netrin-1 receptor[53]. This asymmetric signaling mechanism may be essential or more efficient in the signal transduction from ligands to the receptors. Considering the common dimeric or high-order oligomeric form of surface receptors, the substoichiometric ratio and asymmetric modes of actions between ligands and receptors may represent a paradigm that is more general than is currently known.

## Methods

**Protein expression and purification**. The cDNA of human Ptch1 (Uniprot: Q13635) (residues 1–1305) was cloned into the pCAG vector with an amino-terminal FLAG tag and a carboxy-terminal His$_{10}$ tag. The cDNA of human Shh (Uniprot: Q15465) N-terminal domain (ShhN, residues 1–197) was cloned into the no-tag pCAG vector. A complete list of all primers used in this study has been supplied in Supplementary Table 2. HEK 293F suspension cells (Thermo Fisher Scientific, Cat# R79007) were cultured in Freestyle 293 medium (Thermo Fisher Scientific) at 37 °C supplied with 5% CO$_2$ and 80% humidity. When the HEK 293F cell density reached $2.0 \times 10^6$ cells per ml, the cells were transiently transfected with the expression plasmids and polyethylenimines (PEIs) (Polysciences). For the Ptch1 alone, approximately 1 mg Ptch1 plasmids were pre-mixed with 3 mg PEIs in 50 ml fresh medium for 15–30 min before application. For the Ptch1–ShhN complex, approximately 1 mg Ptch1 and 1 mg ShhN plasmids were pre-mixed with 6 mg PEIs in 50 ml fresh medium for 15–30 min before application. For transfection, 50 ml mixture was added to one-liter cell culture and incubated for 15–30 min. Transfected cells were cultured for 48 h before harvest. For the purification of Ptch1 alone or its complex with ShhN, the HEK 293F cells were collected and resuspended in the buffer containing 25 mM Tris pH 8.0, 150 mM NaCl, and protease inhibitor cocktails (Amresco). After sonication on ice, the membrane fraction was solubilized at 4 °C for 2 h with 1% (w/v) GDN (Anatrace). After

centrifugation at 20,000×g for 1 h, the supernatant was collected and applied to anti-Flag M2 affinity resin (Sigma). The resin was rinsed with the wash buffer (W1 buffer) containing 25 mM Tris pH 8.0, 150 mM NaCl, and 0.02% GDN. The protein was eluted with the W1 buffer plus 200 µg/ml FLAG peptide. The eluent was then applied to the nickel affinity resin (Ni-NTA, Qiagen). After three times of rinsing with W1 buffer plus 20 mM imidazole, the protein was eluted from the nickel resin with W1 buffer plus 250 mM imidazole. The eluent was then concentrated and further purified by size-exclusion chromatography (SEC, Superose® 6 10/300 GL, GE Healthcare) in the buffer containing 25 mM Tris pH 8.0, 150 mM NaCl, and 0.02% GDN. The peak fractions for the oligomeric and monomeric Ptch1 or its complex with ShhN were separately collected.

**Blue Native PAGE (BN-PAGE)**. Chromatographically purified Ptch1 and Ptch1–ShhN$_p$ samples were mixed with 4 × loading buffer and Coomassie G-250 additive and then subjected to 3–12% NativePAGE™ Novex Bis-Tris gel (Invitrogen) for native electrophoresis at 4 °C. The electrophoresis was conducted at 150 V constant for 60 min, and then increase the voltage to 250 V constant for another 75 min. After electrophoresis, the gel was transferred to a container for fixation in fixing solution (40% methanol, 10% acetic acid), and then for staining (0.02% Coomassie R-250 in 30% methanol and 10% acetic acid) and destaining (8% acetic acid). All the procedures were performed according to the manufacturer's protocol. Source data are provided as a Source Data file.

**Glutaraldehyde cross-linking assay**. About 0.15 mg/ml Ptch1 was mixed with glutaraldehyde at indicated concentrations and incubated at 4 °C for 2 h in the buffer containing 25 mM HEPES, 150 mM NaCl and 0.02% GDN. Then the reaction was quenched through the addition of Tris (pH 8.0) to a final concentration of 150 mM and cross-linking result was analysed by SDS-PAGE.

**Analytical ultracentrifugation analyses**. Analytical ultracentrifugation sedimentation velocity (AUC-SV) experiments were performed using a Beckman Coulter XL-I analytical ultracentrifuge equipped with a four-cell An-60 Ti analytical rotor. Four hundred µl protein sample from size-exclusion chromatography in the buffer containing 25 mM Tris pH 8.0, 150 mM NaCl, and 0.02% GDN and 400 µl buffer (25 mM Tris PH 8.0, 150 mM NaCl) were loaded into the sample sector and reference sector separately. The rotor with the cells was pre-scanned at a rotor speed of 70 × g for sample leakage. The run was started once the rotor temperature reached the set point at 20 °C. Sedimentation profiles were recorded by UV detector at 280 nm and Interference laser at 655 nm and scanned every 6 min.

Data were analyzed with GUSSI. The partial specific volume and dn/dc value for the protein are 0.74 cm$^3$/g and 0.1896 cm$^3$/g, respectively, according to the report. The extinction coefficient for Ptch1 is 1.27 L/(g cm), which was calculated with ProtParam on the ExPasy server. For GDN, the partial specific volume was measured by Density Meter as 0.80 cm$^3$/g and dn/dc value was measured by multi-angle light scattering as 0.135 cm$^3$/g.

**Cryo-EM sample preparation and data collection**. The cryo grids were prepared using Thermo Fisher Vitrobot Mark IV. The Quantifoil R1.2/1.3 Cu grids were first glow-discharged with air for 40 s at medium level in Plasma Cleaner (HARRICK PLASMA, PDC-32G-2). Then aliquots of 3.5 µl purified Ptch1–ShhN complex (concentrated to approximately 15 mg/ml) were applied to glow-discharged grids. After being blotted with filter paper for 3.5 s, the grids were plunged into liquid ethane cooled with liquid nitrogen. A total of 4003 micrograph stacks were automatically collected with SerialEM on Titan Krios at 300 kV equipped with K2 Summit direct electron detector (Gatan), Quantum energy filter (Gatan) and Cs corrector (Thermo Fisher), at a nominal magnification of ×105,000 with defocus values from −2.0 µm to −1.2 µm. Each stack was exposed in super-resolution mode for 5.6 s with an exposing time of 0.175 s per frame, resulting in 32 frames per stack. The total dose was about 50 e$^-$/Å$^2$ for each stack. The stacks were motion corrected with MotionCor2[54] and binned twofold, resulting in a pixel size of 1.114 Å/pixel, meanwhile dose weighting was performed[55]. The defocus values were estimated with Gctf[56].

**Cryo-EM data processing**. A total of 1,226,114 particles were automatically picked with RELION 2.0[57]. After 2D classification, a total of 448,682 particles were selected and subject to a guided multi-reference classification procedure. The references, two good and two bad, were generated with limited particles in advance. A total of 266,572 particles selected from multi-references 3D classification were subjected to local search 3D classification with adapted mask on the flexible half (to obtain a complete map) and performed five parallel runs at the same time. Then, a total of 107,265 particles were selected from good classes and subjected to seven parallel runs of local search 3D classification without adapted mask. Two distinct classes were selected, Class I with 39,503 particles and Class II with 25,510 particles, yielding 3D reconstitutions with 6.8 Å and 6.5 Å, respectively. Lastly, ECD masks were applied to increase the local resolution to 4.6 Å and 4.3 Å for these two classes, respectively.

To increase the resolution of stable half, the 266,572 particles selected from multi-references 3D classification were subjected to a global angular search 3D classification with one class and 40 iterations. The outputs of the 30th–40th iterations were subjected to local angular search 3D classification with three classes separately. A total of 171,590 particles were selected by combining the good classes of the local angular search 3D classification, yielding a 3D reconstruction with an overall resolution of 3.6 Å after 3D auto-refinement with an adapted mask on the stable half.

All 2D classification, 3D classification, and 3D auto-refinement were performed with RELION 2.0. Resolutions were estimated with the gold-standard Fourier shell correlation 0.143 criterion[58] with high-resolution noise substitution.

**Model building and refinement**. Firstly, the map at 3.6 Å was used to build a 2:1 complex structure of Ptch1 and ShhN. Two previously reported 1:1 complex structures (PDB code 6DMY and 6D4J) served as initial models to be docked into the map with Chimera, followed by manual adjustment in Coot to generate the final structure. Then, two 2:1 structures were fitted into the maps of Class I or Class II to generate two complex structures at 4:2 stoichiometry.

All structure refinements were carried out by PHENIX[59] in real space with secondary structure and geometry restraints. Overfitting of the models was monitored by refining the model against one of the two independent half maps and testing the refined model against the other map[60].

**Reporting summary**. Further information on research design is available in the Nature Research Reporting Summary linked to this article.

## Data availability

Data supporting the findings of this manuscript are available from the corresponding authors upon reasonable request. A reporting summary for this Article is available as a Supplementary Information file. The source data underlying Fig. 1 and Supplementary Figs. 1b and 8c are provided as a Source Data file. Atomic coordinates and EM density maps of Ptch1–ShhN$_p$ complex (PDB: 6N7G, EMDB: EMD-0355 for 4:2 form I; PDB: 6N7K, EMDB: EMD-0358 for 4:2 form II; and PDB: 6N7H, EMDB: EMD-0356 for mask on 2:1) have been deposited in the Protein Data Bank (http://www.rcsb.org) and the Electron Microscopy Data Bank (https://www.ebi.ac.uk/pdbe/emdb/).

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

## Acknowledgements
We thank Paul Shao for technical support during EM image acquisition. We thank the Princeton Imaging and Analysis Center for providing the cryo-EM facility support. This work was supported in part by an investigator award from the Ara Parseghian Medical Research Foundation (to N.Y.). N.Y. is supported by the Shirley M. Tilghman endowed professorship from Princeton University.

## Author contributions
N.Y. and X.G. conceived the project. X.G., H.Q., P.C., M.H. and S.G. performed the experiments. All authors contributed to data analysis. N.Y. and X.G. wrote the manuscript.

## Additional information

**Competing interests:** The authors declare no competing interests.

