## [Peer Review File · Nature Communications]

Reviewers' Comments:

Reviewer #1:

Remarks to the Author:

Qian et al. present an interesting paper that uses cryo-EM to carefully dissect the interaction of the morphogen Sonic hedgehog (Shh) with its 12-transmembrane receptor Patched (Ptch). Their main finding is that Ptch1 protomers are organized as a dimer of dimers assembled by two distinct binding sites for the protein and the N-terminal palmitate adduct, and possibly a third containing Shh C-cholesterol. As stated by the authors, the aim of this work was to resolve the mystery why Shh (as all other Hedgehog (Hh) family members) undergo unusual N-palmitoylation and C-cholesteroylation, because both lipids firmly associate the proteins with the plasma membrane of the producing cell. The question of how Hhs are nevertheless transported across an epithelial field of cells to act on cells at a distance is currently under intense investigation, and in the past years a number of models have been proposed to explain their Hh solubilization, including trafficking by exosomes or cytonemes or the proteolytic shedding of membrane-associated Hhs (which removes both lipids from the solubilized protein core).

The carefully conducted set of experiments by Qian et al. lead to some plausible and generally well supported conclusions that are in line with two recently published cryo-EM studies of another group on the same interaction (Qi et al., published both in Nature and Science (Qi et al., 2018a; Qi et al., 2018b)), as has been outlined by the authors. It's also an important topic, touching a central issue about the transport and regulation of a primary signaling pathway of relevance to development and disease, and the role of palmitate in the process. It is, however, a complicated story, and one which strongly depends on experimental considerations.

Having said this, my major concern is that both the receptor Ptch and the Shh ligand were expressed in, and purified from the same cell (more resembling a setup to study Shh/Dispatched interactions). For this reason, the observed structures will, by definition, contain one or both lipids. In vitro and in vivo, however, this is never physiological, because Hh ligands and Ptch receptors are produced in different cells, which requires their transport. Indeed, a related published Shh/Ptch structure (also shown by Qian et al.) does not involve lipid interactions with the receptor yet explains most known in vitro and in vivo properties of Hh/Ptch interactions (e.g. that the cholesterol modification is not required for Hh biofunction, but Ca²⁺ complexation is, and that antibodies binding to the Ptch-binding site abolish all Hh biofunction). Therefore, in contrast to the authors assumption that both Hh lipids interact with Ptch, the critical question of how Hhs transport from producing to receiving cells needs to be carefully considered: during Hh transport, both Hh lipids sequester inside of lipoproteins, micelles, exosomes or cytonemes (Bischoff et al., 2013; Gradilla et al., 2014; Panakova et al., 2005; Zeng et al., 2001). How, then, do the authors envision their interaction with Ptch in the absence of detergent? The outcome may have been predetermined by the setup, and may not necessarily reflect the physiological situation.

My second point: the authors suggest a new interaction of Shh C-cholesterol with Ptch1B, but, as shown by many other groups, cholesterol is not essential for morphogen signaling (but rather for its ability to form a proper gradient). What, then, is the physiological relevance of the presented Ptch interaction? The least the authors should do is to test the role of C-cholesterol in vitro or in vivo to confirm its suggested role in signaling and to properly discuss what is already known about its contribution to Hh biofunction.

Likewise, several published studies point out that the undisputed functional importance of palmitate may be indirect (Schurmann et al., 2018) or is only restricted to specific cells or tissues (Lee et al., 2001). This is not discussed. Thus, the tone of many of the results presented by Qian et al. is quite definitive and I am not sure whether this is wholly justified - especially given that experimental outline, as described above, may have predetermined the outcome.

Minor points:

How is Hh-GFP biofunction explained? The large tag disturbs the described cholesterol interactions, yet the fusion protein, if expressed under endogenous control, can fully restore Hh function in a H hull background (in the fly) (Chen et al., 2017).

How does their binding model explain dominant negative activities of non-palmitoylated Hh variants on the endogenous protein (Crozatier et al., 2004; Lee et al., 2001)?

Why did the authors use a non-physiological system. eg. to determine the interaction of non-released Shh in the absence of known release factors, such as Scube2 (Creanga et al., 2012; Tukachinsky et al., 2012)? A setting including relevant factors required for Hh release, and possibly Ptch interaction, would have been a much more relevant one.

All Hhs are known to be released in multimeric form. Multimeric Hh would also cluster Ptch at the cell surface, providing another mechanistic model for Ptch clustering even in the absence of lipids. This should be discussed.

A recent study suggested that palmitoylated Hh peptides are sufficient for Ptc-activation (Tukachinsky et al., 2016). This activation does not require the two additional interactions (Ca²⁺ coordination site, cholesterol) described in this manuscript. How is this explained?

Do the authors have an explanation as to why the C-cholesterol is not present in their structure (but in another previously published one)?

Bischoff, M., Gradilla, A. C., Seijo, I., Andres, G., Rodriguez-Navas, C., Gonzalez-Mendez, L. and Guerrero, I. (2013). Cytosomes are required for the establishment of a normal Hedgehog morphogen gradient in *Drosophila* epithelia. *Nat Cell Biol* 15, 1269-1281.

Chen, W., Huang, H., Hatori, R. and Kornberg, T. B. (2017). Essential basal cytosomes take up Hedgehog in the *Drosophila* wing imaginal disc. *Development* 144, 3134-3144.

Creanga, A., Glenn, T. D., Mann, R. K., Saunders, A. M., Talbot, W. S. and Beachy, P. A. (2012). Scube/You activity mediates release of dually lipid-modified Hedgehog signal in soluble form. *Genes Dev* 26, 1312-1325.

Crozatier, M., Glise, B. and Vincent, A. (2004). Patterns in evolution: veins of the *Drosophila* wing. *Trends Genet* 20, 498-505.

Gradilla, A. C., Gonzalez, E., Seijo, I., Andres, G., Bischoff, M., Gonzalez-Mendez, L., Sanchez, V., Callejo, A., Ibanez, C., Guerra, M., et al. (2014). Exosomes as Hedgehog carriers in cytosome-mediated transport and secretion. *Nat Commun* 5, 5649.

Lee, J. D., Kraus, P., Gaiano, N., Nery, S., Kohtz, J., Fishell, G., Loomis, C. A. and Treisman, J. E. (2001). An acylatable residue of Hedgehog is differentially required in *Drosophila* and mouse limb development. *Dev Biol* 233, 122-136.

Panakova, D., Sprong, H., Marois, E., Thiele, C. and Eaton, S. (2005). Lipoprotein particles are required for Hedgehog and Wingless signalling. *Nature* 435, 58-65.

Qi, X., Schmiede, P., Coutavas, E. and Li, X. (2018a). Two Patched molecules engage distinct sites on Hedgehog yielding a signaling-competent complex. *Science* 362.

Qi, X., Schmiede, P., Coutavas, E., Wang, J. and Li, X. (2018b). Structures of human Patched and its complex with native palmitoylated sonic hedgehog. *Nature*.

Schurmann, S., Steffes, G., Manikowski, D., Kastl, P., Malkus, U., Bandari, S., Ohlig, S., Ortmann, C., Rebollido-Rios, R., Otto, M., et al. (2018). Proteolytic processing of palmitoylated Hedgehog peptides specifies the 3-4 intervein region of the *Drosophila* wing. *Elife* 7.

Tukachinsky, H., Kuzmickas, R. P., Jao, C. Y., Liu, J. and Salic, A. (2012). Dispatched and scube mediate the efficient secretion of the cholesterol-modified hedgehog ligand. *Cell Rep* 2, 308-320.

Tukachinsky, H., Petrov, K., Watanabe, M. and Salic, A. (2016). Mechanism of inhibition of the tumor suppressor Patched by Sonic Hedgehog. *Proc Natl Acad Sci U S A* 113, E5866-E5875.

Zeng, X., Goetz, J. A., Suber, L. M., Scott, W. J., Jr., Schreiner, C. M. and Robbins, D. J. (2001). A

freely diffusible form of Sonic hedgehog mediates long-range signalling. Nature 411, 716-720.

Reviewer #2:

Remarks to the Author:

Qian and colleagues showed a beautiful structure of the 4:2 Ptch1/ShhN complex and performed a detailed molecular analysis of half unit - the unit 2:1 complex. By comparing to the recently published study of the 2:1 complex, they found a likely-additional cholesterol binding moiety. Overall, considering the importance of the hedgehog signaling and how much is unknown about it, I recommend the manuscript for publication. However, I feel that the manuscript should also describe more about newly found 4:2 oligomer complex (as opposed to the 2:1 complex) and discuss about the relevance of oligomerization.

Here are my particular suggestions:

1. I suggest to send the MS to a professional editing service, or at least to get it proofread by a native speaker. A number of grammatical errors can make the MS misleading. Some sentences are too casual (for example 'killer experiment', 'blob', 'cryo-sample') or some are grammatically wrong (an extra EM density was founded, p9) or not objective. Some parts of the results seem to belong to the methods. (Just minor errors to correct here: p15, total dose rate -> total dose, p16 40 iteration -> 40 iterations)

2. What I found intriguing is that the authors found a semi-stable oligomeric form (4 Ptch1: 2 ShhN). This has not been reported before. I suggest that the authors make an emphasis on this aspect of the work and describe the dimer-dimer interface, rather than only comparing the structural similarity to the already published structures. What is the nature of the interactions for tetramerization (dimer-dimer connections?), Is there any insight the authors can learn from the fact that they need the detergent GDN, while digitonin (the one used by the previous publication of 2:1 complex) and other conditions that did not stabilize the tetrameric form? Related to this question, the gel filtration profile (Fig. 1b) seems to be very broad, and some populations of complex found in void. How do these components look under EM or under biophysical analysis (such as analytical centrifuge, native gel, DLS, SLS)? Are there any analyzable components (i.e. oligomers with different stoichiometry) in there? Based on that, can the authors speculate a possibility of clustering the Ptch1/ShhN1 on a membrane surface?

3. The tetramers the author reported look interesting and they were stable enough for a structural analysis, and therefore likely have some relevance. This raises the question how relevant they are in a physiological environment. Can you test mutations in the putative interaction interface (ectodomains) between the dimers to see if tetramerization is completely disrupted?

4. Can the authors also clearly differentiate their findings to the cytoplasmic oligomeric domains? The cytoplasmic oligomerization seems unrelated to the oligomer formation described in this study.

5. It is very interesting that AcrB and Ptch1 behave so different, though there is a high sequence similarity. Can you elaborate the comparison to a molecular level (i.e. amino acid) and describe the difference between the inter-oligomer surface of AcrB and the corresponding parts in Ptch1?

Fig. 1a

I suggest to put a structural illustration of Ptch1 and ShhNn including domains in relation to the extra/intra-cellular environment in 3D based on the previously published structure. This would help to understand what is known for the receptor and the basic structure and it would read nicer at the end of the introduction.

Fig. 5: The authors propose a model in which cholesterol transport is inhibited after ShhNp binding –in the Figure, the red arrows are confusing and the small “X”s indicating that the transport is inhibited are almost not visible. Please revise your figure to make it easier to understand.

Reviewer #3:

Remarks to the Author:

General comments:

The paper submitted by Nieng Yan and Xin Gong proposes that the Hedgehog receptor Ptch1 is organized as a tetramer. They report cryo-EM images showing the presence of 2 dimers of Ptch1, each one in interaction with one monomer of palmitoylated ShhN, however, there is no evidence of contact between the two dimers of Ptch1, and no biochemical experiments supporting the hypothesis that Ptch1 is organized and function as a tetramer. Biochemical data are needed to validate this hypothesis. Without that, authors cannot claim that Ptch1 is a tetramer, and should not write the title “Inhibition of tetrameric Patched1 by Sonic Hedgehog through an asymmetric paradigm”.

The 2:1 Ptch1-ShhNp structure obtained by the authors is comparable to that reported in the Science paper recently published by Qi and co-workers, confirming the asymmetric binding of ShhNp with two Ptch1 monomers even in the absence of the Shh cholesterol moiety, and the palmitoyl binding site. This allows to validate that one ShhN binds to two Ptch1. Authors should mention and discuss that in their discussion. They also confirm the cholesterol efflux path in Ptch1 ECD already published. Scrutinizing the electron density maps of the Ptch1-ShhNn complex published by Qi et al, the authors of the manuscript observed the presence of an extra EM density which could correspond to cholesterol. They propose that this steroid binding pocket could accommodate the cholesterol moiety of native ShhN, which brings new insight into the understanding of Shh binding to Ptch1.

Specific comments:

1. Abstract should be re-written:

- The authors wrote “Inhibition of oligomeric Hh receptor Patched (Ptch1) by the secreted and post-translationally modified ligand Hh relieves suppression of the signaling cascades”. This claim is based on only one paper (Fleet et al 2016) which conclusions are not so clear and affirmative. The sentence should be removed or modified. Same comment for the sentence used in the introduction (page 4) “..., despite Ptch1 was shown to be an oligomer in physiological condition” which is far too affirmative.
- The authors wrote “Ptch1-A binds to ShhNp through the well-characterized Ca²⁺-mediated interface on the globular domain of ShhNp, and Ptch1-B primarily interacts with the N-terminal peptide and the palmitoyl moiety”. This is not new but confirms the observations already published in Science by Qi et al.. Authors inverted the name of the Ptch1 protomers given by Qi et al in the Science paper which is very confusing. Please, use the same name as those firstly published for clarity: Ptch1-A for the Ptch1 in interaction with the palmitoyl moiety of ShhN and Ptch1-B for with the Shh-Ca²⁺ binding site.

2. Figure 1a: the authors show the sequence of WT Ptch1. They should present the construct used with shorten c-terminus, and Flag and His tags in N- and C- terminus respectively.

3. Page 10: the sentence “In contrast, tetramerization of Ptch1 is mediated by the intracellular domains lacking interaction among the TMDs and ECDs” is far too affirmative. This could be discussed in the discussion with biochemical data supporting this hypothesis.

4. Page 10: the authors wrote “The tetrameric assembly of Ptch1 may influence the internalization efficiency, a caveat to be investigated”. This should be discussed in the discussion section and developed further. The authors should discuss how a tetrameric assembly of Ptch1 could influence its internalization.

5. ShhN has been proposed to form a trimer. We could then expect to see on EM grids trimers of

dimers of Ptch1. Is it possible that the absence of the cholesteryl moiety of ShhN prevented the formation of the trimer and the observation of a trimer of dimers of Ptch1? This should be discussed.

Response to reviewers' comments:

Reviewer #1:

This reviewer recognized the significance of our study that touched a central issue about the Hh signaling pathway. But he or she is concerned with our implication for the complicated Hh signaling. We agreed with this reviewer and thought further work are required to fully understand this pathway. Our study has raised some important and new points that could be served as a framework for future investigation of Hh signaling. This reviewer raised three major concerns and several minor comments that are addressed below:

Major concerns:

1. *The receptor Ptch and the Shh ligand were expressed in, and purified from the same cell (more resembling a setup to study Shh/Dispatched interactions). For this reason, the observed structures will, by definition, contain one or both lipids. In vitro and in vivo, however, this is never physiological, because Hh ligands and Ptch receptors are produced in different cells, which requires their transport. In contrast to the authors assumption that both Hh lipids interact with Ptch, the critical question of how Hhs transport from producing to receiving cells needs to be carefully considered: during Hh transport, both Hh lipids sequester inside of lipoproteins, micelles, exosomes or cytonemes. How, then, do the authors envision their interaction with Ptch in the absence of detergent? The outcome may have been predetermined by the setup, and may not necessarily reflect the physiological situation.*

We understand this reviewer's concern. The transport of dual-lipidated Hh is a central question about Hh signaling, which has been intensively investigated and several transport modes have been proposed. But the setup of this study is not to investigate the Hh transport. In order to obtain the Ptch1 and **palmitoylated** ShhN complex to perform structural study, we co-expressed Ptch1 and ShhN in HEK293F cells. This co-expression is not for functional study but just for recombinant protein overexpression.

2. *The authors suggest a new interaction of Shh C-cholesterol with Ptch1-B, but, as shown by many other groups, cholesterol is not essential for morphogen signaling (but rather for its ability to form a proper gradient). What, then, is the physiological relevance of the presented Ptch interaction? The least the authors should do is to test the role of C-cholesterol in vitro or in vivo to confirm its suggested role in signaling and to properly discuss what is already known about its contribution to Hh biofunction.*

We thank this reviewer for this insightful question. Indeed, as shown by many

other groups, cholesterol is not essential for morphogen signaling but rather for its ability to form a proper gradient. However, there are also several conflicting reports concerning the role of cholesterol on the Hh activation, which suggest a role of cholesterylation in decreasing, increasing or no additive effect (Feng et al, Development, 2004; Gallet et al, Development, 2006; Tukachinsky et al, Cell Report, 2012; reviewed in Ciepla et al, Biochemical Society Transactions, 2015). These observations imply the complex effect of Hh cholesterylation, which may be species and tissue-dependent. Our structural analysis suggests a new interaction of ShhN C-cholesterol with Ptch1-B. To support the physiological importance of this presented interaction, we added a sentence in the second paragraph of discussion in the revised manuscript “In line up with this, cholesterol modification enhances the potency of signaling activity of Shh ligand in a Shh Light II cell-based report assay³⁷.” (Page 12)

3. Several published studies point out that the undisputed functional importance of palmitate may be indirect (Schurmann et al., 2018) or is only restricted to specific cells or tissues (Lee et al., 2001). This is not discussed. Thus, the tone of many of the results presented by Qian et al. is quite definitive and I am not sure whether this is wholly justified - especially given that experimental outline, as described above, may have predetermined the outcome.

We thank this reviewer for this critical comment. To make the discussion more comprehensive, we extended the third paragraph of discussion section in the revised manuscript “Another Hh solubilization model is the proteolytic shedding of membrane-associated Hh by a protease/sheddase, which removes both lipid moieties⁴³. In this shedding model, the N-palmitoylation is required for the proteolytic removal during solubilization, which suggested an indirect functional importance of palmitoylation^{43,44}. Furthermore, it has been reported that the palmitoylation of Hh is not absolutely required and only restricted to specific species or tissues for signaling activation^{45,46}. Taken together, these observations imply that the functional importance of lipid modifications on Hh is complicated. Our structural studies just provide a plausible functional mechanism and further investigations are required to resolve the disparate findings.” (Page 12 and 13)

Minor comments:

1. How is Hh-GFP biofunction explained? The large tag disturbs the described cholesterol interactions, yet the fusion protein, if expressed under endogenous control, can fully restore Hh function in a Hh null background (in the fly) (Chen et al., 2017).

We thank this reviewer for this insightful question. A possible explanation for this is that although the cholesterol modification could enhance the potency of Hh ligand, the palmitoylated Hh is sufficient to maintain proper function in vivo.

2. *How does their binding model explain dominant negative activities of non-palmitoylated Hh variants on the endogenous protein (Crozatier et al., 2004; Lee et al., 2001)?*

We thank this reviewer for this insightful question. This dominant negative activities of non-palmitoylated Hh variants could be nicely explained by our binding model -- since the non-palmitoylated Hhs could compete for the endogenous Hhs binding to Ptch1 but the non-palmitoylated Hhs are not sufficient to inhibit Ptch1.

3. *Why did the authors use a non-physiological system to determine the interaction of non-released Shh in the absence of known release factors, such as Scube2 (Creanga et al., 2012; Tukachinsky et al., 2012)? A setting including relevant factors required for Hh release, and possibly Ptch interaction, would have been a much more relevant one.*

As our response to the major concern 1, the setup of this study is not to investigate the Hh release or transport. In order to obtain the Ptch1 and **palmitoylated** ShhN complex to perform structural study, we co-expressed Ptch1 and ShhN in HEK293F cells. This co-expression is not for functional study but just for recombinant protein overexpression.

4. *All Hhs are known to be released in multimeric form. Multimeric Hh would also cluster Ptch at the cell surface, providing another mechanistic model for Ptch clustering even in the absence of lipids. This should be discussed.*

We appreciate this insightful comment. Our biochemical and structural study suggest that Ptch1 organized as a tetramer even in the absence of Hh ligands. Considering the multimeric form of Hh ligands, Ptch indeed could cluster into octamers or even higher order oligomers. The exact oligomeric state of multimeric Hh remains elusive. We therefore preferred not to discuss this aspect in the present study.

5. *A recent study suggested that palmitoylated Hh peptides are sufficient for Ptc-activation (Tukachinsky et al., 2016). This activation does not require the two additional interactions (Ca²⁺ coordination site, cholesterol) described in this manuscript. How is this explained?*

We thank this reviewer for this insightful question. A possible explanation for this is that the palmitoylated Hh peptides can bind with the asymmetric Ptc protomers simultaneously to inhibit Ptc and activate signaling.

6. *Do the authors have an explanation as to why the C-cholesterol is not present in*

their structure (but in another previously published one)?

The C-cholesterol is added by an autoproteolytic reaction catalyzed by the C-terminal half of Hh, but we just used the N-terminal half of Hh for recombinant protein expression. Another previously published structure was reconstituted by mixing with **native ShhN** (which was bought from a company and derived from a full-length construct).

We thank this reviewer for his/her time and constructive comments.

Reviewer #2:

This reviewer fully recognized the significance of our study and recommended the manuscript for publication. He or she raised a few specific suggestions that are addressed below:

1. *I suggest to send the MS to a professional editing service, or at least to get it proofread by a native speaker. A number of grammatical errors can make the MS misleading. Some sentences are too casual (for example 'killer experiment', 'blob', 'cryo-sample') or some are grammatically wrong (an extra EM density was founded, p9) or not objective. Some parts of the results seem to belong to the methods. (Just minor errors to correct here: p15, total dose rate -> total dose, p16 40 iteration -> 40 iterations)*

Point taken. We have carefully polished the text in the revised manuscript.

2. *What I found intriguing is that the authors found a semi-stable oligomeric form (4 Ptch1: 2 ShhN). This has not been reported before. I suggest that the authors make an emphasis on this aspect of the work and describe the dimer-dimer interface, rather than only comparing the structural similarity to the already published structures. What is the nature of the interactions for tetramerization (dimer-dimer connections?), Is there any insight the authors can learn from the fact that they need the detergent GDN, while digitonin (the one used by the previous publication of 2:1 complex) and other conditions that did not stabilize the tetrameric form?*

Related to this question, the gel filtration profile (Fig. 1b) seems to be very broad, and some populations of complex found in void. How do these components look under EM or under biophysical analysis (such as analytical centrifuge, native gel, DLS, SLS)? Are there any analyzable components (i.e. oligomers with different stoichiometry) in there? Based on that, can the authors speculate a possibility of clustering the Ptch1/ShhN1 on a membrane surface?

We thank this reviewer for the insightful questions. As we said in the manuscript, the weak interactions between two dimers lead to structural flexibilities that limited the resolution of the tetramer. The local resolution of the dimer-dimer interface is only 6 Å which is not sufficient for sidechain discussion. We therefore just described the dimer-dimer interface as “The two dimers loosely contact each other through ECD2 in both tetrameric assemblies without any interaction in the TMDs” (Page 7, first paragraph). The resolution is also not sufficient to discuss why GDN works better than digitonin or other conditions for stabilization of the tetrameric form.

This broad gel filtration profile was caused by contaminant proteins. In the revised manuscript, we updated a new gel filtration profile (now as Fig. 1a in

the revised manuscript) and ran native PAGE using all the peak fractions, which support the oligomers with same stoichiometry.

Besides, two more pieces of evidence (glutaraldehyde-mediated crosslinking and sedimentation velocity analytical ultracentrifugation) were added in the revised manuscript (Extended Data Fig. 1d,1e) to support the tetrameric assembly of Ptch1.

3. The tetramers the author reported look interesting and they were stable enough for a structural analysis, and therefore likely have some relevance. This raises the question how relevant they are in a physiological environment. Can you test mutations in the putative interaction interface (ectodomains) between the dimers to see if tetramerization is completely disrupted?

We thank this reviewer for this insightful question. In fact, we have tested several ectodomain mutations that try to disrupt the putative dimer-dimer interface, but all failed. As we said in last response, the local resolution of dimer-dimer interface in ectodomains is only 6 Å, thus the interface mutations we have made may not be totally correct. We therefore didn't include this part of data in our manuscript. This result is easy to understand since the oligomer formation is mainly mediated by the cytoplasmic domains and the dimer-dimer interface in ectodomains is insufficient to disrupt the oligomerization.

4. Can the authors also clearly differentiate their findings to the cytoplasmic oligomeric domains? The cytoplasmic oligomerization seems unrelated to the oligomer formation described in this study.

The oligomer formation described in this study is mainly mediated by the cytoplasmic domains. Previous study has already shown that Ptch1 existed as a monomer in the absence of cytoplasmic domains (Qi et al., Nature, 2018).

5. It is very interesting that AcrB and Ptch1 behave so different, though there is a high sequence similarity. Can you elaborate the comparison to a molecular level (i.e. amino acid) and describe the difference between the inter-oligomer surface of AcrB and the corresponding parts in Ptch1?

We thank this reviewer for this insightful question. The potential evolutionary derivation of various RND transporters, including AcrB and Ptch1, has been nicely discussed in a recently published Ptch1 structure paper (Zhang et al., Cell, 2018). We therefore preferred not to discuss this aspect in our manuscript.

6. Fig. 1a: I suggest to put a structural illustration of Ptch1 and ShhNn including domains in relation to the extra/intra-cellular environment in 3D based on the

previously published structure. This would help to understand what is known for the receptor and the basic structure and it would read nicer at the end of the introduction.

We thank this reviewer for this kind suggestion. We have modified the Fig. 1a (now as Extended Data Fig. 1a in the revised manuscript) according to this reviewer's suggestion.

7. Fig. 5: The authors propose a model in which cholesterol transport is inhibited after ShhNp binding –in the Figure, the red arrows are confusing and the small “X” indicating that the transport is inhibited are almost not visible. Please revise your figure to make it easier to understand.

Point taken. We have modified the Fig. 5 according to this reviewer's suggestion in the revised manuscript.

We thank this reviewer for his/her time and constructive comments.

Reviewer #3:

We thank this reviewer for all the critical comments that are addressed below:

General comments:

The major concern of this reviewer is there is no biochemical experiments supporting Ptch1 is organized and function as a tetramer except structural observation. As we known, seeing is believing. We thought structural observation is one of the most important evidences. Also, we considered this reviewer's concerns and added two pieces of new biochemical data to support the structural observation in the revised manuscript. Please refer to the new Extended Data Fig. 1d, 1e and the corresponding main text for details.

Another general comment is about the comparison between our 2:1 structure and another recently published 2:1 structure. This comparison has already been discussed in Page 9 of our manuscripts as "Structural comparison of the two 2:1 complexes reveals nearly identical architecture, except the two Ptch1* molecules move towards each other relative to the ones in our Ptch1-ShhN_p complex (Extended Data Fig. 5), possibly owing to the lack of stabilization by the intracellular domains in Ptch1*". The Extended Data Fig. 5 (now as Extended Data Fig. 6 in the revised manuscript) shown the structural comparison of these two structures in details.

Specific comments:

1. Abstract should be re-written:

- *The authors wrote "Inhibition of oligomeric Hh receptor Patched (Ptch1) by the secreted and post-translationally modified ligand Hh relieves suppression of the signaling cascades". This claim is based on only one paper (Fleet et al 2016) which conclusions are not so clear and affirmative. The sentence should be removed or modified. Same comment for the sentence used in the introduction (page 4) "..., despite Ptch1 was shown to be an oligomer in physiological condition" which is far too affirmative.*
- *The authors wrote "Ptch1-A binds to ShhN_p through the well-characterized Ca²⁺-mediated interface on the globular domain of ShhN_p, and Ptch1-B primarily interacts with the N-terminal peptide and the palmitoyl moiety". This is not new but confirms the observations already published in Science by Qi et al. Authors inversed the name of the Ptch1 protomers given by Qi et al in the Science paper which is very confusing. Please, use the same name as those firstly published for clarity: Ptch1-A for the Ptch1 in interaction with the palmitoyl moiety of ShhN and Ptch1-B for with the Shh-Ca²⁺ binding site.*

Point taken. We have deleted the "oligomeric" word in abstract and re-wrote the

second paragraph in page 4. We also inversed the name of Ptch1-A and Ptch1-B in the revised manuscript.

2. *Figure 1a: the authors show the sequence of WT Ptch1. They should present the construct used with shorten c-terminus, and Flag and His tags in N- and C-terminus respectively.*

Points taken. We have modified the Fig. 1a (now as Extended Data Fig. 1a in the revised manuscript) according to this reviewer's suggestion.

3. *Page 10: the sentence "In contrast, tetramerization of Ptch1 is mediated by the intracellular domains lacking interaction among the TMDs and ECDs" in far too affirmative. This could be discussed in the discussion with biochemical data supporting this hypothesis.*

We thank this reviewer for this critical comment. According to our structure, inter-protomer interactions among TMDs and ECDs are lacking. The tetramerization of Ptch1 was mainly mediated by the intracellular domains which were not well-resolved in our structure. We have added new biochemical data in Extended Data Fig. 1 to support the tetramerization of Ptch1 in the revised manuscript.

4. *Page 10: the authors wrote "The tetrameric assembly of Ptch1 may influence the internalization efficiency, a caveat to be investigated". This should be discussed in the discussion section and developed further. The authors should discuss how a tetrameric assembly of Ptch1 could influence its internalization.*

Points taken. This part has been moved to the last paragraph of the discussion section and re-written as "...The tetrameric assembly of Ptch1 may increase the internalization efficiency or be prerequisite for internalization, a caveat to be investigated"

5. *ShhN has been proposed to form a trimer. We could then expect to see on EM grids trimers of dimers of Ptch1. Is it possible that the absence of the cholesteryl moiety of ShhN prevented the formation of the trimer and the observation of a trimer of dimers of Ptch1? This should be discussed.*

We thank this reviewer for this insightful question. The ShhN in Qi's paper has both N-terminal palmitoylation and C-terminal cholesterylation, but no trimer or trimer of dimers were observed in their studies. Therefore, this speculation may not be right.

We thank this reviewer for all the constructive comments.

Response to reviewers' comments:

Reviewer #1:

This reviewer recognized the significance of our study. He or she raised three major concerns and several minor comments that are addressed below:

Major concerns:

1. *The receptor Ptch and the Shh ligand were expressed in, and purified from the same cell (more resembling a setup to study Shh/Dispatched interactions). For this reason, the observed structures will, by definition, contain one or both lipids. In vitro and in vivo, however, this is never physiological, because Hh ligands and Ptch receptors are produced in different cells, which requires their transport. In contrast to the authors assumption that both Hh lipids interact with Ptch, the critical question of how Hhs transport from producing to receiving cells needs to be carefully considered: during Hh transport, both Hh lipids sequester inside of lipoproteins, micelles, exosomes or cytonemes. How, then, do the authors envision their interaction with Ptch in the absence of detergent? The outcome may have been predetermined by the setup, and may not necessarily reflect the physiological situation.*

We fully understand this reviewer's concern, but it is beyond the scope of our presented study. Our main aim was not to address the transport of Hhs from producing to receiving cells, but to investigate the mechanism of Ptch1 suppression by Hh.

2. *The authors suggest a new interaction of Shh C-cholesterol with Ptch1-B, but, as shown by many other groups, cholesterol is not essential for morphogen signaling (but rather for its ability to form a proper gradient). What, then, is the physiological relevance of the presented Ptch interaction? The least the authors should do is to test the role of C-cholesterol in vitro or in vivo to confirm its suggested role in signaling and to properly discuss what is already known about its contribution to Hh biofunction.*

We thank this reviewer for this insightful question. In fact, the physiological function of the Shh C-cholesterol is not clearly understood. There was report on enhanced potency of Shh signaling (Tukachinsky et al, Cell Reports, 2012). Our structural observation afforded one plausible explanation for the enhanced activity in the presence of the C-cholesterol. Yet, there were also report on the decreased or no change of the activities. It may depend on the distinct species and tissues examined. We have discussed this aspect in the discussion section of our revised manuscript as "Supporting this structural observation, cholesterol modification was shown to enhance the potency of signaling activity of Shh ligand

in a Shh Light II cell-based report assay³⁷. Nonetheless, the function of the cholesterylation remains to be further investigated. Contradictory observations of decreased, increased, or no change of the Hh activation as a result of the cholesterylation have been reported, probably owing to the distinct species and tissues examined³⁸⁻⁴⁰.”

3. *Several published studies point out that the undisputed functional importance of palmitate may be indirect (Schurmann et al., 2018) or is only restricted to specific cells or tissues (Lee et al., 2001). This is not discussed. Thus, the tone of many of the results presented by Qian et al. is quite definitive and I am not sure whether this is wholly justified - especially given that experimental outline, as described above, may have predetermined the outcome.*

We appreciate this critical comment. Per this reviewer’s suggestion, we cited the related reports and discussed the complexity of Shh signaling as “Another Hh solubilization model involves the proteolytic shedding of membrane-associated Hh by a protease, resulting in removal of both lipid moieties⁴⁶. In this model, the N-palmitoylation is required for the proteolytic removal during solubilization^{46,47}. Adding to these solubilization models, our structures suggest that Ptch1 binding could shield both the hydrophobic lipid moieties of ShhN from the aqueous environment (Fig. 5). It is also noted that the palmitoylation of Hh may not be required, but only restricted to specific species or tissues for signaling activation^{48,49}. Taken together, these observations imply that the functional importance of lipid modifications of Hh is complicated. Our structural studies provide an alternative mechanism that awaits further investigations.”

Minor comments:

1. *How is Hh-GFP biofunction explained? The large tag disturbs the described cholesterol interactions, yet the fusion protein, if expressed under endogenous control, can fully restore Hh function in a Hh null background (in the fly) (Chen et al., 2017).*

A possible explanation for this is that although the cholesterol modification could enhance the potency of Hh ligand, the palmitoylated Hh is sufficient to maintain proper function *in vivo*.

2. *How does their binding model explain dominant negative activities of non-palmitoylated Hh variants on the endogenous protein (Crozier et al., 2004; Lee et al., 2001)?*

The dominant negative activity may result from the competition for binding to Ptch1 oligomer. When Ptch1-G is occupied by the non-palmitoylated Hh, Ptch1-P can still perform its transport activity.

3. *Why did the authors use a non-physiological system to determine the interaction of*

non-released Shh in the absence of known release factors, such as Scube2 (Creanga et al., 2012; Tukachinsky et al., 2012)? A setting including relevant factors required for Hh release, and possibly Ptch interaction, would have been a much more relevant one.

As our response to the major concern 1, the setup of this study is not to investigate the Hh release or transport. The co-expression of Ptch1 and ShhN in HEK293F cells is not for functional study but just for recombinant protein overexpression and purification.

4. All Hhs are known to be released in multimeric form. Multimeric Hh would also cluster Ptch at the cell surface, providing another mechanistic model for Ptch clustering even in the absence of lipids. This should be discussed.

We appreciate this insightful comment. We have added one sentence in the last paragraph of discussion “Considering the oligomerization of Hh ligands, this could result in even higher-order clustering of Ptch1 at the cell surface⁵⁰” to discuss this point.

5. A recent study suggested that palmitoylated Hh peptides are sufficient for Ptc-activation (Tukachinsky et al., 2016). This activation does not require the two additional interactions (Ca²⁺ coordination site, cholesterol) described in this manuscript. How is this explained?

It is as expected, because there is no steric hindrance in the binding of palmitoylated Hh peptides to every protomer in the oligomeric Ptch1 at a 1:1 stoichiometric ratio. Hence, the palmitoylated N-terminal peptides of Hh should be able to completely suppress Ptch1 when added at sufficiently high concentration.

6. Do the authors have an explanation as to why the C-cholesterol is not present in their structure (but in another previously published one)?

The C-cholesterol is added by an autoproteolytic reaction catalyzed by the C-terminal half of Hh, but we just used the N-terminal half of Hh for recombinant protein expression. Another previously published structure was reconstituted by mixing with **native ShhN** (which was bought from a company and derived from a full-length construct).

We thank this reviewer for his/her time and constructive comments.

Reviewer #2:

This reviewer fully recognized the significance of our study and recommended publication of the manuscript. He or she raised a few specific suggestions that are addressed below:

1. *I suggest to send the MS to a professional editing service, or at least to get it proofread by a native speaker. A number of grammatical errors can make the MS misleading. Some sentences are too casual (for example 'killer experiment', 'blob', 'cryo-sample') or some are grammatically wrong (an extra EM density was founded, p9) or not objective. Some parts of the results seem to belong to the methods. (Just minor errors to correct here: p15, total dose rate -> total dose, p16 40 iteration -> 40 iterations)*

Point taken. We have carefully polished the text in the revised manuscript.

2. *What I found intriguing is that the authors found a semi-stable oligomeric form (4 Ptch1: 2 ShhN). This has not been reported before. I suggest that the authors make an emphasis on this aspect of the work and describe the dimer-dimer interface, rather than only comparing the structural similarity to the already published structures. What is the nature of the interactions for tetramerization (dimer-dimer connections?), Is there any insight the authors can learn from the fact that they need the detergent GDN, while digitonin (the one used by the previous publication of 2:1 complex) and other conditions that did not stabilize the tetrameric form?*

Related to this question, the gel filtration profile (Fig. 1b) seems to be very broad, and some populations of complex found in void. How do these components look under EM or under biophysical analysis (such as analytical centrifuge, native gel, DLS, SLS)? Are there any analyzable components (i.e. oligomers with different stoichiometry) in there? Based on that, can the authors speculate a possibility of clustering the Ptch1/ShhN1 on a membrane surface?

We thank this reviewer for the insightful questions. As suggested by this reviewer, we have added a new Extended Data Fig in the revised manuscript to describe the dimer-dimer interface (Extended Data Fig. 8) and extended the description about the tetrameric assembly as “In contrast, the tetrameric assembly of Ptch1 only contains limited interaction between the ECDs of the dimer of dimers (Extended Data Figs. 7, 8a, 8b). A variant containing mutations ($\Delta 893-899/T903A/K904A/Q905A/N1000Q$) that alter the residues mediating the limited dimer-dimer interface on the extracellular side failed to disrupt the tetrameric assembly of Ptch1 alone or in complex with ShhN_p (Extended Data Fig. 8c). The result was as expected because the tetrameric assembly is mainly mediated by the intracellular domains, deletion of which resulted in the monomeric of Ptch1 both *in vivo* and *in vitro*^{21,23}.” (Page 11)

This broad gel filtration profile was caused by contaminant proteins. In the revised manuscript, we updated a new gel filtration profile (now as Fig. 1a in the revised manuscript) and ran native PAGE using all the peak fractions, which support the oligomers with same stoichiometry.

Besides, two more pieces of evidence (glutaraldehyde-mediated crosslinking and sedimentation velocity analytical ultracentrifugation) were added in the revised manuscript (Extended Data Fig. 1c,1d) to support the tetrameric assembly of Ptch1.

3. The tetramers the author reported look interesting and they were stable enough for a structural analysis, and therefore likely have some relevance. This raises the question how relevant they are in a physiological environment. Can you test mutations in the putative interaction interface (ectodomains) between the dimers to see if tetramerization is completely disrupted?

We thank this reviewer for this insightful question. As we said in last response, a mutation (Δ 893-899/T903A/K904A/Q905A/N1000Q) that designed to disrupt the dimer-dimer interface didn't affect the tetrameric assembly of Ptch1 both alone and in complex with ShhN_p (as Extended Data Fig. 8c in the revised manuscript). This result is easy to understand since the tetramer formation is mainly mediated by the cytoplasmic domains, and the dimer-dimer interface mutation in ectodomains is insufficient to disrupt the tetramerization.

4. Can the authors also clearly differentiate their findings to the cytoplasmic oligomeric domains? The cytoplasmic oligomerization seems unrelated to the oligomer formation described in this study.

The oligomer formation described in this study is mainly mediated by the cytoplasmic domains. Previous study has already shown that Ptch1 existed as a monomer in the absence of cytoplasmic domains (Qi et al., Nature, 2018).

5. It is very interesting that AcrB and Ptch1 behave so different, though there is a high sequence similarity. Can you elaborate the comparison to a molecular level (i.e. amino acid) and describe the difference between the inter-oligomer surface of AcrB and the corresponding parts in Ptch1?

We thank this reviewer for this insightful question. The potential evolutionary derivation of various RND transporters, including AcrB and Ptch1, has been nicely discussed in a recently published Ptch1 structure paper (Zhang et al., Cell, 2018). We therefore preferred not to discuss this aspect in our manuscript.

6. *Fig. 1a: I suggest to put a structural illustration of Ptch1 and ShhNn including domains in relation to the extra/intra-cellular environment in 3D based on the previously published structure. This would help to understand what is known for the receptor and the basic structure and it would read nicer at the end of the introduction.*

We thank this reviewer for this kind suggestion. We have modified the Fig. 1a (now as Extended Data Fig. 1a in the revised manuscript) according to this reviewer's suggestion.

7. *Fig. 5: The authors propose a model in which cholesterol transport is inhibited after ShhNp binding –in the Figure, the red arrows are confusing and the small “X” indicating that the transport is inhibited are almost not visible. Please revise your figure to make it easier to understand.*

Point taken. We have modified the Fig. 5 according to this reviewer's suggestion in the revised manuscript.

We thank this reviewer for his/her time and constructive comments.

Reviewer #3:

We thank this reviewer for all his/her critical comments that have helped the improvement of our manuscript.

General comments:

1. *The major concern of this reviewer is there is no biochemical experiments supporting Ptch1 is organized and function as a tetramer except structural observation.*

To address this general concern, we added two pieces of new biochemical data to support the structural observation in the revised manuscript. Please refer to the new Extended Data Fig. 1c, 1d and the corresponding main text for details. We would also like to point out that it is very easy to disrupt an oligomer during membrane protein extraction and purification. However, it is extremely rare, if it ever happens, that proteins homo-oligomerize in vitro but not in vivo.

2. *Another general comment is about the comparison between our 2:1 structure and another recently published 2:1 structure.*

This comparison has already been discussed on Page 9 of our manuscripts as “Structural comparison of the two 2:1 complexes reveals nearly identical architecture, except the two Ptch1* molecules move towards each other relative to the ones in our Ptch1-ShhN_p complex (Extended Data Fig. 6), possibly owing to the lack of stabilization by the intracellular domains in Ptch1*²”. The Extended Data Fig. 6 shown the structural comparison of these two structures in details.

Specific comments:

1. *Abstract should be re-written:*

- *The authors wrote “Inhibition of oligomeric Hh receptor Patched (Ptch1) by the secreted and post-translationally modified ligand Hh relieves suppression of the signaling cascades”. This claim is based on only one paper (Fleet et al 2016) which conclusions are not so clear and affirmative. The sentence should be removed or modified. Same comment for the sentence used in the introduction (page 4) “..., despite Ptch1 was shown to be an oligomer in physiological condition” which is far too affirmative.*
- *The authors wrote “Ptch1-A binds to ShhN_p through the well-characterized Ca²⁺-mediated interface on the globular domain of ShhN_p, and Ptch1-B primarily interacts with the N-terminal peptide and the palmitoyl moiety”. This is not new but confirms the observations already published in Science by Qi et al. Authors inversed the name of the Ptch1 protomers given by Qi et al in the Science paper*

which is very confusing. Please, use the same name as those firstly published for clarity: Ptch1-A for the Ptch1 in interaction with the palmitoyl moiety of ShhN and Ptch1-B for with the Shh-Ca²⁺ binding site.

We thank this reviewer for the critical comments. We have deleted the word “oligomeric” from the abstract and re-wrote the second paragraph on page 4. To make the illustration clearer, we re-named the two Ptch1 protomers as Ptch1-G (G for interaction with the globular domain of ShhN) and Ptch1-P (P for interaction with the palmitoylate and peptide of ShhN) in the revised manuscript.

2. Figure 1a: the authors show the sequence of WT Ptch1. They should present the construct used with shorten c-terminus, and Flag and His tags in N- and C-terminus respectively.

Points taken. We have modified the Fig. 1a (now as Extended Data Fig. 1a in the revised manuscript) according to this reviewer’s suggestion.

3. Page 10: the sentence “In contrast, tetramerization of Ptch1 is mediated by the intracellular domains lacking interaction among the TMDs and ECDs” in far too affirmative. This could be discussed in the discussion with biochemical data supporting this hypothesis.

Point taken. We have toned down the description and rewrote the paragraph on Page 11 “In contrast, the tetrameric assembly of Ptch1 only contains limited interaction between the ECDs of the dimer of dimers (Extended Data Figs. 7, 8a, 8b). A variant containing mutations (Δ 893-899/T903A/K904A/Q905A/N1000Q) that alter the residues mediating the limited dimer-dimer interface on the extracellular side failed to disrupt the tetrameric assembly of Ptch1 alone or in complex with ShhN_p (Extended Data Fig. 8c). The result was as expected because the tetrameric assembly is mainly mediated by the intracellular domains, deletion of which resulted in the monomeric of Ptch1 both *in vivo* and *in vitro*^{21,23}.” We also added new biochemical data in Extended Data Fig. 1 to support the tetramerization of Ptch1 in the revised manuscript.

4. Page 10: the authors wrote “The tetrameric assembly of Ptch1 may influence the internalization efficiency, a caveat to be investigated”. This should be discussed in the discussion section and developed further. The authors should discuss how a tetrameric assembly of Ptch1 could influence its internalization.

Points taken. This part has been moved to the discussion section (last paragraph) and re-written as “...The tetrameric assembly of Ptch1 may increase the internalization efficiency or be prerequisite for internalization, a caveat to be investigated”

5. ShhN has been proposed to form a trimer. We could then expect to see on EM

gives trimers of dimers of Ptch1. Is it possible that the absence of the cholesteryl moiety of ShhN prevented the formation of the trimer and the observation of a trimer of dimers of Ptch1? This should be discussed.

We thank this reviewer for this insightful question. The ShhN in Qi's paper has both N-terminal palmitoylation and C-terminal cholesterylation, but no trimer or trimer of dimers were observed in their studies.

We thank this reviewer for all the constructive comments.

Reviewers' Comments:

Reviewer #1:

Remarks to the Author:

In their revised paper, Qian et al. address some but not all of the concerns raised by this reviewer. Most importantly, I believe that the authors conclusion (that the Shh receptor Patched (Ptch) is organized as a dimer of dimers assembled by three distinct binding sites for the Shh protein and the N-terminal palmitate/C-terminal cholesterol) does not necessarily describe the in vivo situation.

Let me explain my point. As outlined previously, my major concern is that both the receptor Ptch and the Shh ligand were expressed in, and purified from the membrane of the same producing cell. Therefore, the observed structures will, by definition, contain the Shh lipid (Here, as in a previous study conducted by the same authors, the experimental setup determined the outcome). Moreover, the analyzed Shh variant contained only the N-palmitate but no cholesterol as a result of deleting the C-terminal autoprocessing domain. Such a protein is not known in nature, but can only be artificially produced. This raises the possibility that the observed interactions are of limited biological relevance.

Indeed, several lipidated and non-lipidated Shh proteins were recently analyzed by Cryo-EM (Gong et al., 2018; Qi et al., 2018a; Qi et al., 2018b). The different observed structures and interactions strongly imply that the structural outcome depends on the Shh "input". Because this is so, an essential question that needs to be answered in this study (and for all previous studies) is: Which of these tested Shh variants represents the physiologically relevant form "seen" by the receptor Ptch? It is known that the 12-pass transmembrane protein Dispatched (Disp) is absolutely essential for Shh release from the membrane of the producing cell, and that only Disp-released Shh will be encountered by Ptch receptors on receiving cells. Only this form should therefore be analyzed. Yet, it is not clear whether Disp-released Shh contains one, two or no lipids. The authors used a Shh protein variant that has not gone through Disp-dependent release (instead it was extracted from the membrane of producing cells, skipping the essential step of Disp-mediated Shh release). This is another experimental detail that worries me.

Having said this, I am not happy with the authors reply that this concern "is beyond the scope of our presented study". To the contrary, the question of whether the analyzed Shh variant is biologically relevant determines the value of the obtained structure. The same applies to dual-lipidated Shh analyzed in this and a previous study: While this variant represents the membrane-associated protein on producing cells, it is not clear whether Shh stays unchanged during its Disp-mediated release. Several reports indicate that this may not be the case, because Disp-overexpression can rescue the knockdown of proteases that release truncated Shh from its lipidated terminal peptides (Damhofer et al., 2015) and bioactive non-lipidated Shh can be detected in vitro (Ohlig et al., 2011) and in vivo (Palm et al., 2013). Although the authors now touch these concerns in their discussion section, the major conclusion of the study, as it still stands, just represents one possibility out of many, and its biological relevance is unclear.

The least the authors should do is to much more clearly point this out. Better, still, would be to test experimentally whether C-cholesterol contributes critically to Shh/Ptch interactions. This is also important because the authors decided to analyze an artificial protein lacking the sterol in their present study, as outlined above.

Minor points:

In their reply (point 5), the authors point out that palmitoylated Shh peptides can completely suppress Ptch-activity at sufficiently high concentrations, supporting a role of N-palmitate in Ptch activity suppression. However, this is not in line with the observations that mutating the Zn²⁺

coordination site leads to structural shifts that abolish all Hh biofunction (supporting Ptch-binding at the Ca²⁺ interface, but insufficient lipid functions) (Day et al., 1999; Fuse et al., 1999).

Line 284: Reference #44 should be cited together with references #38-40, as unchanged activity of non-cholesteroylated Hh is described.

References

- Damhofer, H., Veenstra, V. L., Tol, J. A., van Laarhoven, H. W., Medema, J. P. and Bijlsma, M. F. (2015). Blocking Hedgehog release from pancreatic cancer cells increases paracrine signaling potency. *J Cell Sci* 128, 129-139.
- Day, E. S., Wen, D., Garber, E. A., Hong, J., Avedissian, L. S., Rayhorn, P., Shen, W., Zeng, C., Bailey, V. R., Reilly, J. O., et al. (1999). Zinc-dependent structural stability of human Sonic hedgehog. *Biochemistry* 38, 14868-14880.
- Fuse, N., Maiti, T., Wang, B., Porter, J. A., Hall, T. M., Leahy, D. J. and Beachy, P. A. (1999). Sonic hedgehog protein signals not as a hydrolytic enzyme but as an apparent ligand for patched. *Proc Natl Acad Sci U S A* 96, 10992-10999.
- Gong, X., Qian, H., Cao, P., Zhao, X., Zhou, Q., Lei, J. and Yan, N. (2018). Structural basis for the recognition of Sonic Hedgehog by human Patched1. *Science*.
- Ohlig, S., Farshi, P., Pickhinke, U., van den Boom, J., Hoing, S., Jakushev, S., Hoffmann, D., Dreier, R., Scholer, H. R., Dierker, T., et al. (2011). Sonic hedgehog shedding results in functional activation of the solubilized protein. *Dev Cell* 20, 764-774.
- Palm, W., Swierczynska, M. M., Kumari, V., Ehrhart-Bornstein, M., Bornstein, S. R. and Eaton, S. (2013). Secretion and signaling activities of lipoprotein-associated hedgehog and non-sterol-modified hedgehog in flies and mammals. *PLoS Biol* 11, e1001505.
- Qi, X., Schmiede, P., Coutavas, E. and Li, X. (2018a). Two Patched molecules engage distinct sites on Hedgehog yielding a signaling-competent complex. *Science* 362.
- Qi, X., Schmiede, P., Coutavas, E., Wang, J. and Li, X. (2018b). Structures of human Patched and its complex with native palmitoylated sonic hedgehog. *Nature*.

Reviewer #2:

Remarks to the Author:

The authors answered the criticisms sufficiently and I am happy to recommend for acceptance for the publication.

Reviewer #3:

Remarks to the Author:

Overall, the authors have taken my suggestions into account and have greatly improved the manuscript, which I believe can now be published in *Nature Communications*.

Following please find our response to reviewer #1's criticism:

As outlined previously, my major concern is that both the receptor Ptch and the Shh ligand were expressed in, and purified from the membrane of the same producing cell. Therefore, the observed structures will, by definition, contain the Shh lipid (Here, as in a previous study conducted by the same authors, the experimental setup determined the outcome). Moreover, the analyzed Shh variant contained only the N-palmitate but no cholesterol as a result of deleting the C-terminal autoprocessing domain. Such a protein is not known in nature, but can only be artificially produced. This raises the possibility that the observed interactions are of limited biological relevance.

Please be noted that the interface between our N-terminus palmitoylated and C-terminus unmodified ShhNp with dimeric Patched is **identical** to that in the fully-modified Shh bound to dimeric Patched by Qi et al (Science, 2018). If ours is artifact, so would be theirs. In that study, a purified, fully modified ShhN was added to purified Patched. Based on this reviewer's idea, this would be a native one. **The similarity between the two structures except for the oligomeric states of Patched, which has nothing to do with the C-terminal modification of ShhN, supported the physiological relevance of our present structural observations.**

Indeed, several lipidated and non-lipidated Shh proteins were recently analyzed by Cryo-EM (Gong et al., 2018; Qi et al., 2018a; Qi et al., 2018b). The different observed structures and interactions strongly imply that the structural outcome depends on the Shh "input".

In fact, there was **NO** real difference among these structures. They all reflect pieces of a big picture. **All the structures in the three referred papers are consistent with each other.**

Because this is so, an essential question that needs to be answered in this study (and for all previous studies) is: Which of these tested Shh variants represents the physiologically relevant form "seen" by the receptor Ptch? It is known that the 12-pass transmembrane protein Dispatched (Disp) is absolutely essential for Shh release from the membrane of the producing cell, and that only Disp-released Shh will be encountered by Ptch receptors on receiving cells. Only this form should therefore be analyzed. Yet, it is not clear whether Disp-released Shh contains one, two or no lipids. The authors used a Shh protein variant that has not gone through Disp-dependent release (instead it was extracted from the membrane of producing cells, skipping the essential

step of Disp-mediated Shh release). This is another experimental detail that worries me.

We respectfully disagree with the reviewer for this specific criticism and we really don't regard this question to be essential in our study. For structural biology, construct optimization, including internal or terminal truncations, point mutations, and various fusion proteins or affinity tags, has been a common strategy for structural determination. The validation of a structure is more meaningful than the materials used per se. As mentioned above, the ShhN used by Qi *et al* contains lipidations on both ends and that structure is consistent with ours. It is simply because of the difference of the C-terminal modification, did we realize that they failed to build structural model for this moiety. The two structures, ours and Qi's, actually represent perfect control to each other.

Having said this, I am not happy with the authors reply that this concern "is beyond the scope of our presented study". To the contrary, the question of whether the analyzed Shh variant is biologically relevant determines the value of the obtained structure. The same applies to dual-lipidated Shh analyzed in this and a previous study: While this variant represents the membrane-associated protein on producing cells, it is not clear whether Shh stays unchanged during its Disp-mediated release. Several reports indicate that this may not be the case, because Disp-overexpression can rescue the knockdown of proteases that release truncated Shh from its lipidated terminal peptides (Damhofer et al., 2015) and bioactive non-lipidated Shh can be detected in vitro (Ohlig et al., 2011) and in vivo (Palm et al., 2013). Although the authors now touch these concerns in their discussion section, the major conclusion of the study, as it still stands, just represents one possibility out of many, and its biological relevance is unclear. The least the authors should do is to much more clearly point this out. Better, still, would be to test experimentally whether C-cholesterol contributes critically to Shh/Ptch interactions. This is also important because the authors decided to analyze an artificial protein lacking the sterol in their present study, as outlined above.

We are sorry for the less careful wording by "*is beyond the scope of our presented study*" in our previous response. What we meant was the following: there have been reports on the various effects of the C-terminal cholesterylation, positive, negative, and no effect. Assuming all these results are reproducible, it suggests that the effect of this modification is system-dependent. Whatever results we obtain would only give rise to one of

these three conclusions, which won't exclude the correctness of the other contradictory results. It is impractical for any group to perform experiments in all kinds of experimental systems. We therefore replied that it was beyond the scope of our present study.

In fact, the similar structures of the Shh-Patched complex with or without the C-terminal cholesteryl already provided an important clue to understanding the functional role of this modification.

- 1. The cholesteryl is not essential for complex formation as both the globular domain and the palmitoylated N-terminal peptide are sufficient to ensure complex formation.**
- 2. The cholesteryl group is inserted into the extracellular pocket of one Patched protomer, further inhibiting potential cholesterol transport of that protomer.**

Therefore, it is reasonable to predict that the effect of this modification is concentration-dependent. When the concentration of both Shh and Patched are high enough, the effect of cholesterylation is undetectable. But when the concentration is low, this binding may strengthen the binding between the ligand and the receptor. It is known that the secreted Shh ligand diffuses to form a gradient. Therefore, the effect of this modification may totally depend on the context, a caveat to be examined by developmental biologists.

On the other hand, the globular domain of ShhN moves slightly toward Patched-G in our structure in the absence of the C-terminal modification. In fact, because of the very short linker between the globular domain and the cholesteryl moiety, binding of cholesterol into the pocket of Patched-P may drag the globular domain away from Patched-N. Therefore, the distance between the two Patched protomers may determine the effect of the cholesterol as well. Because there is no contact between the dimer on the extracellular and transmembrane domains, it is possible that the dimeric assembly may be regulated by some intracellular factors, hence offering another tier of difference for the effect of the modified Shh on Patched function. This is something we

are currently investigating, but it requires a lot of experiments that cannot be completed in the near future.